# Uncovering origins of heterogeneous superconductivity in La$_3$Ni$_2$O$_7$

S. V. Mandyam[1,6], E. Wang[1,6], Z. Wang[1,6], B. Chen[1,6], N. C. Jayarama[2], A. Gupta[1], E. A. Riesel[3], V. I. Levitas[4], C. R. Laumann[1,2,5✉] & N. Y. Yao[1✉]

The family of nickelate superconductors have long been explored as analogues of the high-temperature cuprates[1–6]. Nonetheless, the recent discovery that certain stoichiometric nickelates superconduct up to high critical temperatures ($T_c$) under pressure came as a surprise[7–13]. The mechanisms underlying the superconducting state remain experimentally unclear. Apart from the practical challenges posed by working in a high-pressure environment, typical samples exhibit anomalously weak diamagnetic responses, which have been conjectured to reflect inhomogeneous 'filamentary' superconducting states[7,9,14–17]. Here we perform wide-field, high-pressure, optically detected magnetic resonance spectroscopy to image the local diamagnetic responses of as-grown La$_3$Ni$_2$O$_7$ samples in situ, using nitrogen vacancy quantum sensors embedded in the diamond anvil cell[18–23]. These maps confirm marked inhomogeneity of the functional superconducting responses at the few μm scale. By spatially correlating the diamagnetic Meissner response with both the local tensorial stress environment, also imaged in situ, and stoichiometric composition, we show the dominant mechanisms suppressing and enhancing superconductivity. Our wide-field technique simultaneously provides a broad view of sample behaviour and excellent local sensitivity, enabling the rapid construction of multi-parameter phase diagrams from the local structure–function correlations observed at the sub-μm pixel scale.

Nickelate materials play an important part in the decades-long quest to probe, understand and enhance high-temperature (high-$T_c$) superconductivity[24–29]. They provide a fresh perspective on the unconventional physics of the cuprates[1–6], realizing similar electronic and structural motifs in a wholly different material platform. For example, the seminal observation of superconductivity in thin-film, square planar nickelates, which are isovalent to the cuprates, hints at a more universal relationship between superconductivity and orbital filling[4,25,26,30].

More recently, a tremendous amount of excitement has focused on the discovery of superconductivity in the bulk nickelate, La$_3$Ni$_2$O$_7$, with a critical temperature above the boiling point of liquid nitrogen[7]. This discovery challenges the nascent model connecting nickelate and cuprate physics: La$_3$Ni$_2$O$_7$ exhibits an electronic configuration that is distinct from the cuprates and superconducts only at high pressures ≥10 GPa (refs. 8–14). These distinctions suggest the potential to both broaden and deepen our understanding of high-$T_c$ superconductivity. However, realizing this potential comes with its own set of obstacles.

Perhaps the most important, from a practical perspective, is the presence of substantial variations in the superconducting properties measured across different La$_3$Ni$_2$O$_7$ samples[7,9,14–17]. Even when superconductivity is observed, variations exist in the magnitude and sharpness of the drop in resistance, the transition temperature, the local diamagnetic response[22,23], the onset pressure and the characteristics of the normal state[7,14,16]. Moreover, measurements of La$_3$Ni$_2$O$_7$ have

also observed very low superconducting volume fractions, leading the superconductivity to be dubbed 'filamentary'[10,15]; these observations complicate our understanding of both the nature of nickelate superconductivity and the underlying connection to the cuprates.

Marked progress has been made towards a qualitative understanding of the microscopic origins of the above variations, with invocations to local inhomogeneity in the chemistry[14,31–33], structure[15,34] and stress environment[9,10,16]. However, many questions remain—for example, whether one of the purported forms of inhomogeneity dominates the superconducting response of La$_3$Ni$_2$O$_7$. More quantitatively, it remains unclear whether there is an interplay between these inhomogeneities and whether we can identify the associated parameter space hosting superconductivity. The fact that these questions are fundamentally related to the role of local inhomogeneities makes them extremely difficult to answer. Ideally, we would want to spatially correlate the local superconducting properties of La$_3$Ni$_2$O$_7$ with maps of the various types of inhomogeneity. The difficulty of this pursuit is markedly exacerbated by the high-pressure setting, in which the local imaging of functional material properties remains a perennial challenge.

Here, we take a crucial step towards addressing this challenge. With sub-μm spatial resolution, we directly correlate local regions of superconductivity in La$_3$Ni$_2$O$_7$ with spatial maps of both the stress environment and the chemical composition (Fig. 1). This is achieved through our multimodal, wide-field sensing techniques (Fig. 1a), which extend

[1]Department of Physics, Harvard University, Cambridge, MA, USA. [2]Department of Physics, Boston University, Boston, MA, USA. [3]Department of Chemistry, Massachusetts Institute of Technology, Cambridge, MA, USA. [4]Departments of Aerospace and Mechanical Engineering, Iowa State University, Ames, IA, USA. [5]Max-Planck-Institut für Physik Komplexer Systeme, Dresden, Germany. [6]These authors contributed equally: S. V. Mandyam, E. Wang, Z. Wang, B. Chen. ✉e-mail: claumann@bu.edu; nyao@fas.harvard.edu

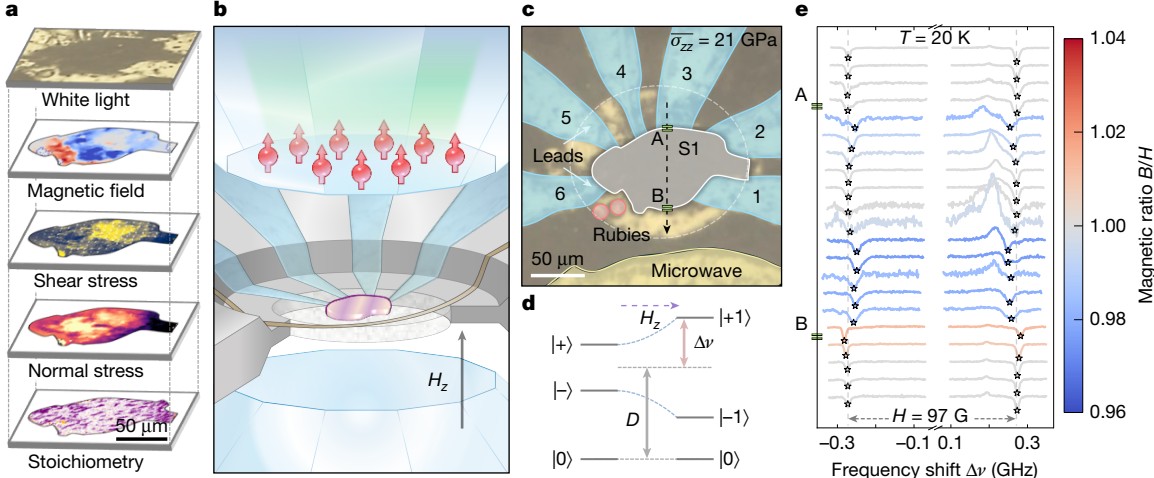

**Fig. 1 | Micrometre-scale structure–function mapping at high pressure in an NV-DAC. a**, The NV-DAC enables sub-µm-scale imaging of functional magnetic responses of samples at high pressure. Correlating these maps with the local stress environment and chemical composition yields a wealth of structure–function information. **b**, Schematic of the sample loading. The top anvil contains a layer of NV centres about 500 nm below the culet surface. We note that there is an approximately 3° misalignment between the culet normal and the [111] NV symmetry axis, which is discussed in the Methods. **c**, White-light image of sample S1 with false colour overlays obtained at 21 GPa. A crystal of $La_3Ni_2O_7$ (grey) is embedded in NaCl as a pressure medium (tan) within an insulating cBN gasket (dark). **d**, Schematic of the spin $S = 1$ sub-levels of the electronic ground state of the NV centre as a function of an external magnetic field $H_Z$. **e**, The ODMR spectra obtained at regularly spaced pixels along the line-cut indicated after ZFC to 20 K and turning on a magnetic field of $H = 97$ G. Away from the sample boundaries (points A and B), the measured splitting $\Delta\nu \approx 0.27$ GHz is consistent with a local $B = 97$ G, whereas above the sample, there are regions in which $\Delta\nu$ is both smaller (magnetic suppression) and larger (enhancement) than normal state expectations. Spectra are scaled to have uniform peak contrast but are otherwise unprocessed. We observe inverted positive contrast peaks, which extend our ability to measure both magnetic fields and traction to higher pressures (see the Methods for details).

beyond recent confocal studies of high-pressure nickelates[22,23]. Our main results are threefold. First, by using diamond anvil cells (DACs) instrumented with a shallow layer of nitrogen-vacancy (NV) colour centres[18–21], we perform wide-field, high-pressure, optically detected magnetic resonance spectroscopy (ODMR) to image the local diamagnetic response—associated with the superconducting Meissner effect—in three separate $La_3Ni_2O_7$ samples: S1 (Fig. 2), S2 (Fig. 3) and S3 (Fig. 5). We note that these samples are carefully chosen to exhibit differing degrees of chemical homogeneity as measured using energy-dispersive X-ray spectroscopy (EDX). Crucially, apart from NV-based measurements of the diamagnetism, we simultaneously measure the transport behaviour of the samples, observing a drop in resistance concomitant with the onset of the Meissner effect (Fig. 2c,d). The proximity of our NV centres to the $La_3Ni_2O_7$ sample yields excellent magnetic-field sensitivity, which enables the first observation of flux trapping in the nickelates. Moreover, we find a direct correlation between those regions of the sample exhibiting diamagnetism and those that trap flux (Fig. 2e).

Second, using a complementary modality of the NV sensors, we image the three components of the local stress tensor ($\overleftrightarrow{\sigma}$) that define the so-called traction vector[35]: $\mathbf{f} = \{\sigma_{XZ}, \sigma_{YZ}, \sigma_{ZZ}\}$. Crucially, **f** is continuous across the diamond–sample interface, providing a map of the local stresses experienced by the $La_3Ni_2O_7$ sample. The traction vector consists of two physically distinct contributions: (1) the normal stress $\sigma_{ZZ}$ and (2) the shear stress vector $\boldsymbol{\tau} = \{\sigma_{XZ}, \sigma_{YZ}\}$ with magnitude $\tau = |\boldsymbol{\tau}| = \sqrt{\sigma_{XZ}^2 + \sigma_{YZ}^2}$, which arises owing to contact friction between the sample and the diamond. Near the critical pressure, we observe the first signatures of superconductivity confined to localized regions of the sample only where the normal stress, $\sigma_{ZZ}$, is sufficiently large (Fig. 4b,c). On further compression, these superconducting regions spread to encompass substantial fractions of the entire sample (Fig. 2a). Perhaps most intriguingly, our spatial maps of the shear stress magnitude, $\tau$, yield the following conclusion: above a critical shear of approximately 2 GPa, the superconducting behaviour of $La_3Ni_2O_7$ quenches (Fig. 4d,e). Although conventional phase diagrams of nickelate superconductivity depict pressure as a single axis with discrete steps[5,7,9,14,16], our ability to measure the local stress environment allows us to access a more refined and continuous 'pressure' axis (Fig. 4g,h). By decomposing this pressure axis into its normal and shear components, we arrive at a complex three-dimensional (3D) superconducting phase diagram for $La_3Ni_2O_7$ as a function of $\{T, \sigma_{ZZ}, \tau\}$ (Fig. 4f).

Finally, by choosing to work with samples exhibiting noticeable chemical inhomogeneities (S3), we observe the following behaviour: local pockets of superconductivity exhibiting sharp diamagnetic transitions even in the absence of an overall drop in sample resistance (Fig. 5b). Crucially, these superconducting pockets overlap with regions of optimal stoichiometry (Fig. 5e).

## NV spectroscopy of $La_3Ni_2O_7$

Our experiments are performed on three independent floating-zone-synthesized single crystals (S1, S2 and S3) of $La_3Ni_2O_7$ (ref. 36) mounted within DACs and connected to transport leads (Fig. 1b,c). The top anvil in each DAC is a type Ib (111)-cut diamond, implanted with a thin layer of NV centres around 500 nm below the culet surface at a density of approximately 1 ppm (refs. 19,21) (Methods). The workhorse of our investigation is the ability of NV to sense the local magnetic field and stress environment. Both magnetic-field- and stress-induced perturbations to the energies of the spin sub-levels of NV can be directly read out by ODMR spectroscopy[18]: owing to differences in the fluorescence of the NV spin sub-levels, a microwave field swept through resonance leads to a characteristic dip in the measured photoluminescence (Methods). By imaging this fluorescence onto a CCD, our experiments perform wide-field ODMR spectroscopy and resolve the effects of both perturbations across the entire high-pressure sample chamber (with sub-µm spatial resolution) in a single shot[37]. We observe that samples do not experience a single 'pressure', and so report the averaged normal stress of the sample, $\overline{\sigma_{ZZ}}$. We find that this metric agrees well with the sample chamber pressure as measured by ruby (Methods).

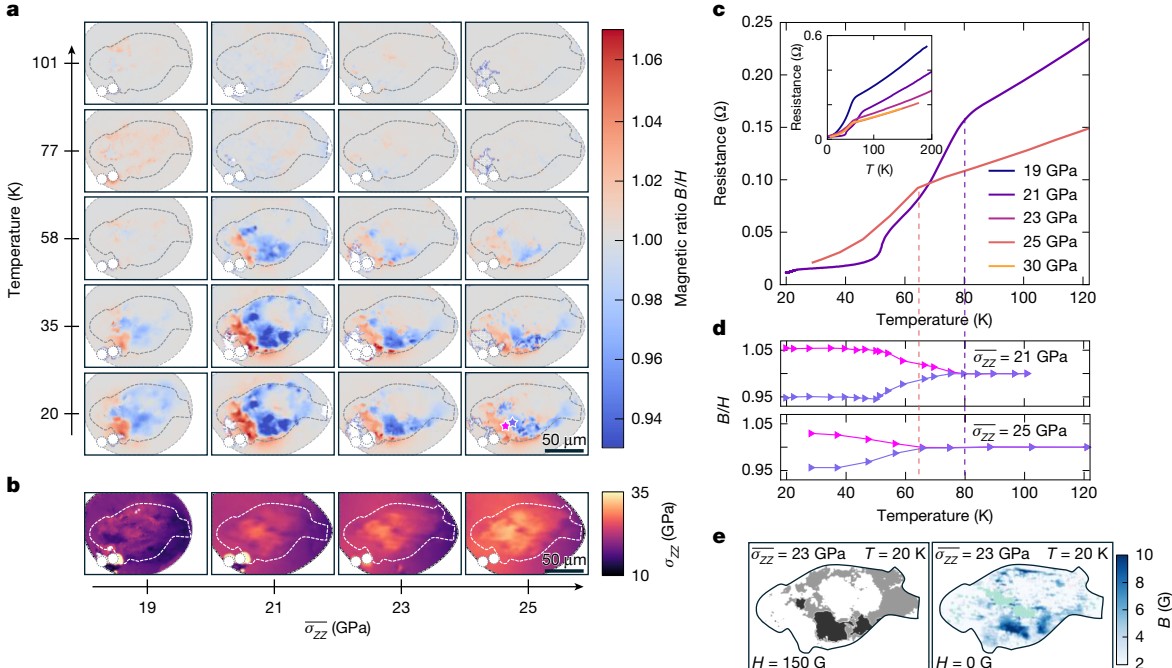

**Fig. 2 | Imaging local superconductivity and flux trapping. a**, Sub-μm diffraction-limited maps of sample S1, showing the magnetic field $B$ obtained after ZFC to 20 K, turning on $H \approx 100$ G and then field warming (FW). The magnetic ratio $s \equiv B/H$ above the sample deviates from 1 below a dome in the $\overline{\sigma_{ZZ}}$, $T$ plane, although clear spatial inhomogeneities exist. In particular, the asymmetric magnetic textures are suggestive of the Wohlleben effect (Methods). **b**, Corresponding $\sigma_{ZZ}$ maps for these stress points, taken at 150 K. **c,d**, Simultaneously measured resistance at $\overline{\sigma_{ZZ}} = 21$ GPa and 25 GPa, with the ZFC-FW magnetic response of two spatial points marked on **a** (pink and purple

stars, bottom right). Kinks in the magnetic response correspond to kinks in resistance at corresponding stress points. **e**, Spatial regions of strongest ZFC diamagnetic response (left) at $\overline{\sigma_{ZZ}} = 23$ GPa correspond to regions with the most remnant magnetic flux (right) trapped after field cooling (FC) at $H = 150$ G and quenching to $H = 0$ G. For the left panel, grey regions correspond to $0.97 \leq B/H < 1$, whereas black regions correspond to $0.85 \leq B/H < 0.97$. For the right panel, green regions correspond to ODMR spectra with NV resonances that were unable to be resolved.

## Transport, Meissner effect and flux trapping

To investigate the local diamagnetic response of La$_3$Ni$_2$O$_7$, we perform zero-field cooling (ZFC), field warming and field cooling studies (for protocol details, see Methods). In all cases, we apply a uniform external magnetic field, $H$, along $\hat{Z}$, and then use NV centres to measure the local magnetic field, $B$, above the sample (Fig. 1b). A diamagnetic sample response, consistent with the superconducting Meissner effect, manifests as a particularly simple expectation: for NVs directly above a diamagnetic region, we expect $s \equiv \frac{B}{H} < 1$, corresponding to a local suppression of the magnetic field; for NVs near the edge of a diamagnetic region, we expect $s > 1$, corresponding to the bunching of expelled magnetic field lines; finally, for NVs away from the sample, we simply expect to measure the applied external field and thus $s \approx 1$ (refs. 38–40) (see Methods for benchmark).

Let us begin by considering the ZFC experimental sequence. At low stresses, $\overline{\sigma_{ZZ}} \lesssim 15$ GPa, for all three La$_3$Ni$_2$O$_7$ samples, we observe non-superconducting transport behaviour and ODMR spectra consistent with $s = 1$ throughout the entire high-pressure chamber (Extended Data Figs. 2 and 3). Focusing on S1, as one increases to higher stresses, the NV centres directly above the sample (Fig. 1b) begin to exhibit ODMR spectra with $s \neq 1$. As an example, Fig. 1e shows the ODMR spectra (at $\overline{\sigma_{ZZ}} = 21$ GPa) obtained along a line-cut that spans the high-pressure chamber. For NVs outside of the sample region (solid white outline, Fig. 1c), the ODMR resonances are split by precisely the frequency expected for our external applied field of $H = 97$ G (yielding $s \approx 1$). For NVs inside the sample region, three distinct types of contiguous ODMR behaviour are observed: (1) regions in which the ODMR resonances are split by less than expected from the applied field (Fig. 1e, blue curves, with $s < 1$); (2) regions in which the ODMR resonances are split by more

than expected from the applied field (Fig. 1e, red curves, with $s > 1$); and (3) regions in which the ODMR resonances are consistent with the applied field (Fig. 1e, grey curves, with $s \approx 1$).

Although we have initially focused on an ODMR line-cut, our wide-field NV microscopy allows us to directly create two-dimensional (2D) maps of the local diamagnetic response (as characterized by $s$). Figure 2a shows a series of these maps as a function of both normal stress and temperature; here, the temperature values refer to a field warming experiment performed after the previous experimental sequence (that is, ZFC and then apply $H$), whereas the normal stress refers to the average of $\sigma_{ZZ}$ shown in Fig. 2b. At relatively low normal stresses and high temperatures, we do not observe any signatures of the La$_3$Ni$_2$O$_7$ sample in the maps of $s$, because $s \approx 1$ uniformly throughout the high-pressure chamber. At low temperatures and near the optimal normal stress (around $\overline{\sigma_{ZZ}} = 21$ GPa), we find a pronounced region (Fig. 2a, blue) of magnetic-field suppression exhibiting an annular shape; on the left side of this suppression region, we also observe a pronounced region (Fig. 2a, red) of magnetic-field enhancement. Both of these features are co-localized with the sample (black dashed line), indicating that they arise from the diamagnetic response of La$_3$Ni$_2$O$_7$. As we continue to increase the normal stress, two effects are seen: first, the magnitude of the diamagnetic response of the sample becomes weaker, and second, the spatial extent of the regions exhibiting this response also decreases (Fig. 2a). Taken together, these data (Fig. 2a) evince a particularly elegant visualization of the superconducting dome of La$_3$Ni$_2$O$_7$ in the stress–temperature phase diagram.

More quantitatively, Fig. 2d presents the magnetic ratio $s$ as a function of temperature for two specific locations within the sample; one point is chosen to be in a region exhibiting magnetic-field suppression (Fig. 2a, purple star, bottom right), whereas the other is chosen from

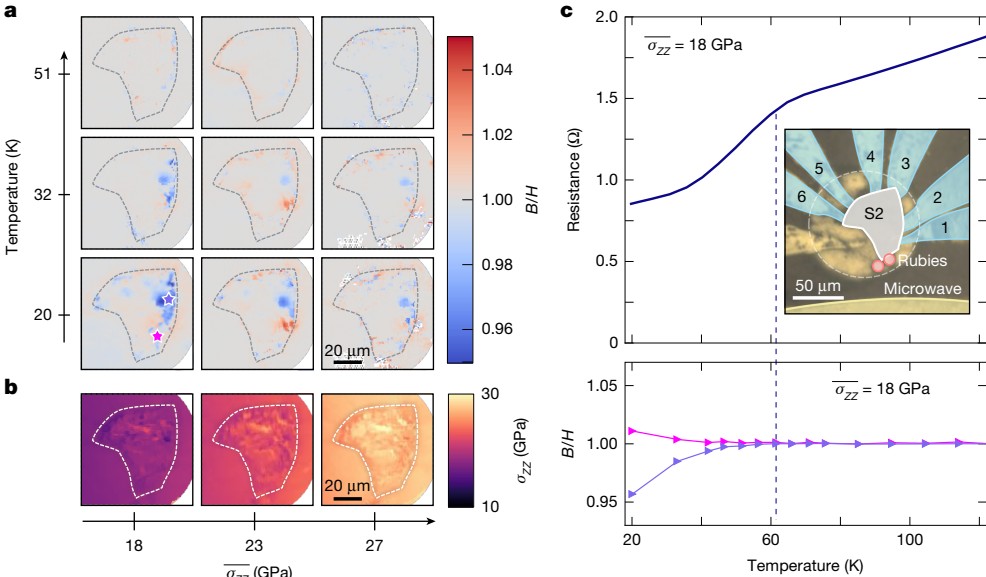

**Fig. 3 | Local superconductivity and normal stress maps. a,b,** Maps of the ZFC-FW magnetic response (**a**) and normal stress maps of sample S2 (**b**) analogous to those in Fig. 2 for sample S1. **c,** Correlation of the resistive transition (above) with the magnetic response (below) at one stress point. Sample S2 shows qualitatively similar diamagnetic and resistive responses to those of sample S1 in the $\overline{\sigma_{ZZ}} - T$ plane but in a smaller spatial volume and with a weaker local *s* response.

a region exhibiting magnetic-field enhancement (Fig. 2a, pink star, bottom right). For $\overline{\sigma_{ZZ}} = 21$ GPa, both points exhibit a plateau in *s* (at $s \approx 0.95$ (purple) and $s \approx 1.05$ (pink)) as the temperature is increased from 20 K to 50 K, before gradually converging to $s = 1$ at higher temperatures; the temperature at which the two points meet naturally defines a transition temperature of $T_c \approx 80$ K. Nearly identical behaviour is observed for $\overline{\sigma_{ZZ}} = 25$ GPa, except that the diamagnetism converges to $s = 1$ at a lower critical temperature, $T_c \approx 65$ K. Both of these critical temperatures obtained from the diamagnetic response of the sample are in perfect agreement with the temperature at which the resistance exhibits its first drop (Fig. 2c). This concurrence strongly suggests that the observed local diamagnetism (Fig. 2a) originates from the superconducting Meissner effect. To further solidify our findings, we perform the same set of measurements on a second sample, S2, as shown in Fig. 3. We find analogous behaviour both in the local maps of diamagnetism (albeit with a significantly smaller region exhibiting $s < 1$) as a function of stress and temperature (Fig. 3a) and in the concurrence between transport and diamagnetism (Fig. 3c).

Returning to S1, we now leverage the enhanced magnetic-field sensitivity of our proximal NV sensors to directly image flux trapping in $La_3Ni_2O_7$. In particular, we cool the sample from room temperature down to 20 K in an applied field of $H = 150$ G. Then we turn off the magnetic field and perform wide-field ODMR spectroscopy at $H = 0$ G. As shown in Fig. 2e, many points (in the sample region) exhibit an ODMR spectrum consistent with a remnant magnetic field of up to around 10 G. These regions of trapped flux coincide with the regions of Meissner-induced diamagnetism (Fig. 2e); more quantitatively, the regions exhibiting the largest diamagnetic response also trap the largest amount of flux.

## The role of normal and shear stresses

Our ability to locally determine the sample regions exhibiting a diamagnetic Meissner response opens the door to the following possibility: directly identifying the cause of the observed μm-scale spatial variations in superconductivity. Let us begin by exploring the role of the local stress environment of the samples. That there is a superconducting dome in the stress–temperature phase diagram (Figs. 2 and 3a)

immediately yields the following prediction—if there exists substantial heterogeneity in the local normal stress experienced by the sample, then near the phase transition, we expect certain regions of the sample to go superconducting first. This simple expectation is borne out by the data. In particular, Fig. 4c shows the local normal stress experienced by sample S1 when $\overline{\sigma_{ZZ}} = 16$ GPa; we observe marked variations of the normal stress ranging from 13 GPa to 18 GPa, and we identify a high-normal-stress region of the sample (Fig. 4c, dashed black oval) exhibiting $\sigma_{ZZ} \gtrsim 17$ GPa. The sample begins to exhibit its first signatures of superconductivity in an area that is fully contained within this high-stress region (Fig. 4b, dashed black oval). These observations highlight the limitations intrinsic to assuming a uniform stress environment and attributing superconductivity to a single bulk pressure.

Perhaps more importantly, at each temperature, we can correlate, pixel by pixel, the superconducting response of the sample, *s*, with its local $\sigma_{ZZ}$. This has two important consequences: (1) it essentially renders the stress axis of our superconducting phase diagram continuous because every pressure point contains a broad distribution of local normal stresses and (2) it enables a more refined stress–temperature phase diagram in which each point is characterized by a quantitative measure of the superconducting response of the sample, *s*. Combining the data across all three samples yields the temperature/normal-stress phase diagram shown in Fig. 4g, which is in excellent agreement with previous studies[7,9,14,16].

We now turn to data at higher stresses near the optimum. Recall that for sample S1 (at $\overline{\sigma_{ZZ}} = 21$ GPa), the diamagnetic suppression at the lowest temperatures exhibits a distinct annular shape, with a large region near the centre of the sample showing little or no Meissner response (Fig. 4d). We might naturally wonder whether this region is again correlated with spatial variations of the normal stress; we find that this is not the case (Fig. 2a,b). Instead, we observe a striking correlation between the image of the diamagnetic response of the sample and a map of the local shear stress magnitude $\tau$ (Fig. 4e). This correlation immediately yields the following conjecture, namely, that superconductivity in $La_3Ni_2O_7$ is eliminated by the presence of sufficiently large shear stresses. To further explore this conjecture, we again leverage our ability to correlate, pixel by pixel, the superconducting response of the sample with its local shear. In particular, as shown in

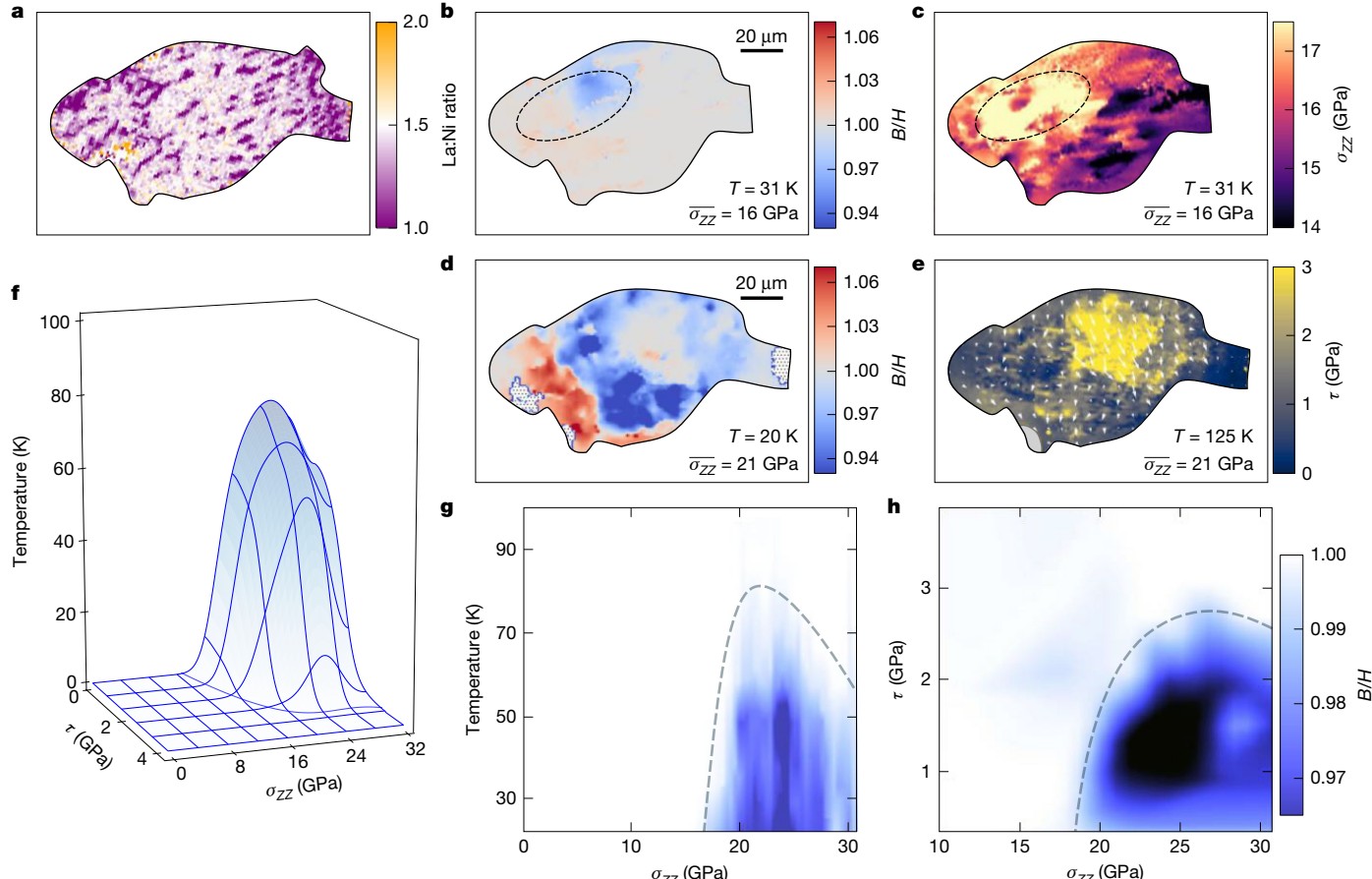

**Fig. 4 | Multimodal correlations, superconducting phase diagram and the role of shear stresses. a**, The local surface stoichiometry of sample S1 obtained by EDX shows regular Ni rich inclusions (purple) at the several μm scale as well as smaller regions of enhanced La (orange). This chemical texture alone does not explain the bulk features observed in the ZFC magnetic response. For example, at the lowest stress point, $\overline{\sigma_{ZZ}}$ = 16 GPa, where S1 exhibits diamagnetic response. **b**, The region of the sample that goes superconducting turns out to be locally at a higher normal stress $\sigma_{ZZ} \approx 17.5$ GPa than the mean shown in **c**. **e**, The shear stress vector **τ** with components $(\sigma_{XZ}, \sigma_{YZ})$ (white arrows) across the culet boundary at optimal mean stress, $\overline{\sigma_{ZZ}}$ = 21 GPa. Colour indicates the magnitude of the shear stress vector **τ**. The region of high shear correlates strongly with the hole in the annular region of diamagnetic response in the

corresponding ZFC-FW magnetic response shown in panel **d**. The grey region indicates overlap with the ruby pellet, which prevents the accurate extraction of the local shear. **f**, A 3D phase diagram of the superconducting response as a function of temperature, normal stress and shear stress. Data are extrapolated down to zero on all axes using a spline fit (Methods). Note that there is no data extrapolation for the phase diagrams shown in the subsequent panels **g** and **h**. **g**, The superconducting dome in the temperature, normal stress plane extracted from pixel-registered local responses across samples S1–S3. The dashed line corresponds to a guide to the eye delineating the superconducting region. **h**, A low-temperature projection of the phase diagram in the normal and shear stress plane. The dashed line provides a guide to the eye delineating the superconducting region.

Fig. 4h, combining the data across all samples, we generate a superconducting phase diagram in the normal-stress/shear-stress plane, uncovering a superconducting dome with a critical shear stress of approximately 2 GPa (Methods). More generally, the pixel-level data allows us to refine the conventional 2D pressure–temperature phase diagram of the superconductivity of $La_3Ni_2O_7$ into a 3D phase diagram as a function of temperature, normal stress and shear stress (Fig. 4f).

## The role of chemical composition

Finally, we investigate the role of local chemical composition in controlling nickelate superconductivity. It is well known that multiple distinct Ruddlesden-Popper phases[41], distinguished by different La:Ni stoichiometric ratios, compete during the crystal growth process[26,27,36,42]. Even from the same $La_3Ni_2O_7$ crystal, we obtain samples exhibiting significant variations in chemical homogeneity as determined by EDX spectroscopy. To this end, we now turn to sample S3, which is chosen specifically for its strong spatial variations in La:Ni stoichiometry. In particular, unlike sample S1 (Fig. 4a) and S2 (Extended Data Fig. 4),

which exhibit La:Ni stoichiometries that are relatively homogeneous across the sample, S3 exhibits two pronounced stripe regions with an La:Ni ratio far from the expected optimum of 3:2 (Fig. 5d). This variation in chemistry leads to an immediate impact on the transport behaviour of the sample, and we do not observe any drop in resistance (Fig. 5b, top) even near the optimal stress, $\overline{\sigma_{ZZ}}$ = 21 GPa. Nevertheless, at the lowest temperatures ($T$ = 9 K), ODMR spectroscopy (on ZFC and applying an external field) shows pockets of weak diamagnetic suppression (Fig. 5f); as before, this suppression vanishes as we increase the temperature above $T_c \approx 60$ K (Fig. 5b, bottom). Overlapping the EDX and ODMR data shows the following qualitative conclusion: sample regions exhibiting an La:Ni ratio away from 3:2 do not exhibit a diamagnetic response (Fig. 5e). Perhaps more surprisingly, nearly all regions with the correct stoichiometry do seem to exhibit a weak diamagnetic response. More quantitatively, by registering each pixel of the EDX with its associated value of the local superconducting suppression, we demonstrate that a clear optimum in $s$ emerges at La:Ni = 3:2 (Fig. 5c); this pixel-by-pixel registration also enables the construction of a stoichiometry–temperature phase diagram as shown in Extended Data Fig. 4 (Methods).

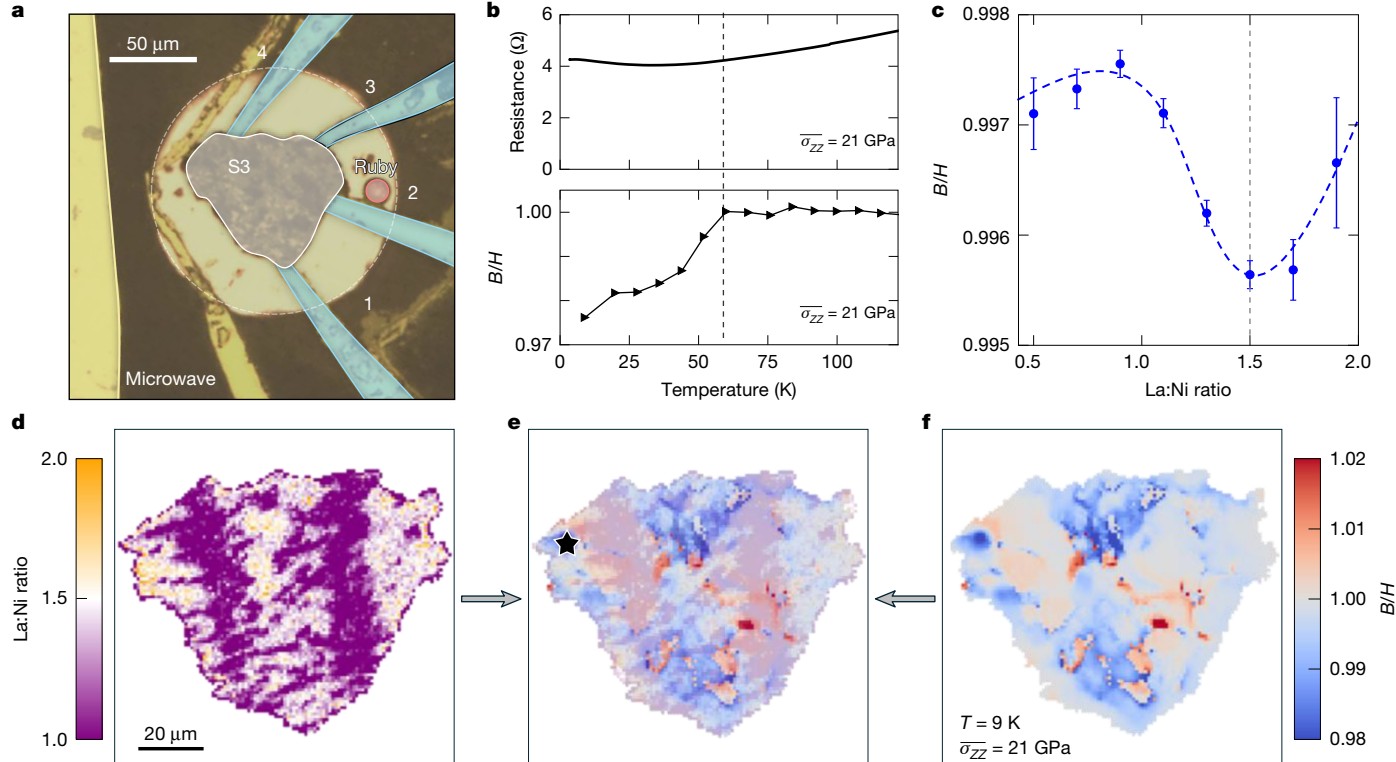

**Fig. 5 | Correlating local superconductivity with chemistry. a**, Optical image of S3 (grey) with transport leads (blue), microwave antenna (yellow) and ruby (red). **b**, Global four-terminal resistance (above) shows no visible kinks even as the local diamagnetic response (black star in **e**) turns on below 60 K (below). **c**, The local ZFC diamagnetic response ($B/H$) at 9 K shows a sharp peak in which the ratio of La:Ni obtained by EDX is in the expected 3:2 ratio of stoichiometric $La_3Ni_2O_7$. The dashed line serves as a guide to the eye. **d**, The spatial map of the La:Ni ratio shows two large stripes of enhanced Ni, which cut across S3. These correlate strongly with regions that fail to show a magnetic response when overlaid (**e**) with the magnetic map obtained after ZFC at 9 K as shown in **f**.

## Discussion and conclusion

Our results are important for both the understanding of nickelate superconductivity and high-pressure metrology in general. Traditionally, inhomogeneity in samples and sample loadings has rightfully been seen as a notable source of systematic error in high-pressure science, and a correspondingly great effort has been expended on achieving homogeneous materials and hydrostatic pressure environments[43,44]. With the ability to locally image both functional material properties and the tensorial stress environment, we may instead view a single inhomogeneous sample as a collection of many μm-scale samples that can be simultaneously characterized to construct multi-parameter phase diagrams. Rather than a source of error to be mitigated, inhomogeneity becomes a resource to be mined. This perspective lies behind our selection of the particular $La_3Ni_2O_7$ samples that were measured and enabled our observation of the role of shear in suppressing and eliminating the superconducting state. We note that this complements recent observations of the importance of in-plane stress in epitaxially strained thin films[42,45].

Our work opens the door to several intriguing future directions. First, by resolving the distribution of stresses within a sample, we effectively convert pressure into a continuous tuning knob. Second, our magnetic maps show regions of paramagnetic response that turn on below $T_c$ (for example, Fig. 2a). One such possibility is the so-called Wohlleben effect first observed in granular cuprate superconductors and taken as early indirect evidence for exotic pairing symmetry[46–48] (Methods). Third, although our approach has identified several mechanisms underlying the heterogeneity of $La_3Ni_2O_7$ superconductivity, much variability remains to be explained (for example, sample S2). Other mechanisms, such as oxygen stoichiometry[31–33,45] and layer stacking variations[34], may be interrogated by comparing NV-DAC magnetic maps with imaging techniques such as nanoscale secondary ion mass spectrometry[49] and μm resolution synchrotron X-ray diffraction[50]. Fourth, understanding the microscopic mechanism by which shear stress destroys superconductivity in $La_3Ni_2O_7$ is an intriguing open question. Example potential mechanisms include plastic deformations or a shear distortion of the angle of the Ni–O bond, which controls the coupling between bilayers.

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

# Methods

## NV ground state Hamiltonian

Each NV centre features an $S = 1$ electronic spin ground state, described by the Hamiltonian $H_0 = D_{gs}S_Z^2$, where $\hat{Z}$ corresponds to the quantization axis along the NV axis[51,52]. In the absence of external perturbations, the zero-field splitting, $D_{gs} = (2\pi) \times 2.87$ GHz, captures the energy difference between the $m_s = 0\rangle$ spin sub-level and the two degenerate $m_s = \pm 1\rangle$ sub-levels (Fig. 1d); initialization to the $m_s = 0\rangle$ state is achieved by optical pumping using a 532 nm laser[40,53–55].

Both magnetic field and stress alter the relative energies of the spin sub-levels (Fig. 1d). In particular, magnetic fields $\mathbf{B}$ couple through the Zeeman Hamiltonian $H_B = \mathbf{B} \cdot \mathbf{S}$ (refs. 56–59), where $\gamma$ is the NV spin gyromagnetic ratio, and stress $\overset{\leftrightarrow}{\sigma}$, which is a rank-2 tensor with six independent elements, couples through the stress Hamiltonian, $H_\sigma = \Pi_Z S_Z^2 + \Pi_X(S_Y^2 - S_X^2) + \Pi_Y(S_X S_Y + S_Y S_X)$ (refs. 18,60,61). Here, $(\Pi_X, \Pi_Y, \Pi_Z)$ are functions of the stress tensor that transform according to the irreducible representations of the NV point group: $\Pi_X = -(b + c)(\sigma_{YY} - \sigma_{XX}) + (\sqrt{2}b - \frac{c}{\sqrt{2}})(2\sigma_{XZ})$, $\Pi_Y = -(b + c)(2\sigma_{XY}) + (\sqrt{2}b - \frac{c}{\sqrt{2}})(2\sigma_{YZ})$ and $\Pi_Z = (a_1 - a_2)(\sigma_{xx} + \sigma_{YY}) + (a_1 + 2a_2)\sigma_{ZZ}$, where $\{a_1, a_2, b, c\}$ correspond to the stress susceptibilities of the NV centre[18,62].

## Measurement protocols

As superconductors support metastable currents, they often show hysteretic behaviour. Thus, it is crucial to keep track of the thermo-dynamic history of the sample. We use the following measurement protocols, which always begin at $T > T_c$. Each of these steps is also shown graphically in Extended Data Fig. 1b.

**Diamagnetism.** For the $B/H$ measurements performed in both the main text and in the Extended Data, we use the following protocol:
1. ZFC: we begin by cooling the sample below the transition temperature $T_c$ without any applied field ($H = 0$). Electrical transport is recorded during this step.
2. The field $H$ is turned on. We typically choose fields $H$ of the order 100–150 G, applied along $\hat{z}$.
3. Field warming: the temperature is then raised in steps, and the transport and magnetometry measurements are taken simultaneously during this warming procedure.

**Flux trapping.** We use two different protocols to measure flux trapping. The first protocol, detailed below, is used in the flux trapping study of La$_3$Ni$_2$O$_7$ (Fig. 2e and Extended Data Fig. 7b):
1. ZFC: we perform ZFC to 20 K, and then perform NV magnetometry to collect a reference ZFC spectrum.
2. Field cooling: a 150 G field is turned on at 150 K, where the sample is in the normal state. Then, we cool below $T_c$ to 20 K.
3. The field is turned off, and NV magnetometry is performed to collect the field-cooled spectrum.

This method is further detailed below in the 'Flux trapping and analysis' section.

In the next section, we perform benchmarking experiments on Bi$_2$Sr$_2$CaCu$_2$O$_{8+\delta}$. For this material, we use a slightly different flux trapping protocol, in which we force the flux into the system while it is already in the superconducting state:
1. ZFC: the sample is cooled below $T_c$ without any applied field.
2. The field is turned on. The field should be at a magnitude great enough that it begins to penetrate through the superconductor.
3. The field is turned off, and NV magnetometry is performed. Again, without the presence of the external field, any remnant non-zero magnetic field must be present as a result of trapped flux.

**Field cooling.** The last measurement protocol that we use is the field cooling protocol (Extended Data Fig. 6d):

1. Similar to the first flux trapping protocol, the field is turned on while the sample is in the normal state. Here, the sample is cooled in a sequence of temperature increments, during which NV magnetometry is performed. Here, due to the flux trapped as the system transitions into the mixed phase, we expect a different $B/H$ response than in the ZFC-FW protocol.

## Notation

We reserve uppercase indices $I, J$ and coordinates $X, Y, Z$ for the lab frame defined by the culet and imaging system. More precisely, $\hat{Z}$ is perpendicular to the culet and $\hat{X}, \hat{Y}$ are defined by the orientation of the CCD imaging plane, see Extended Data Fig. 1a. This is the most natural frame for mechanical considerations.

The response of the NV centre is most naturally described in the frame adapted to its symmetry axis, as is conventional[18]. As there are four NV orientations, there are four such frames, which we label as $g = 0, 1, 2, 3$, when it is necessary to disambiguate (Extended Data Fig. 8b). We adopt lowercase indices $i, j$ and coordinates $x, y, z$ for NV frames. Thus, the NV$^{(g)}$ group has symmetry axes $\hat{z}^{(g)}$. We define the global NV frame to be that of the $g = 0$ group and typically suppress the $g$ label for quantities in that frame.

Each NV group contributes two resonance peaks to the ODMR spectrum observed in a pixel. We define $D_g$ to be the frequency of the midpoint of the two peaks associated with group $g$ and $E_g$ to be their half-difference. We note $E = \Delta\nu$, as defined in Fig. 1d, and we use the two representations interchangeably.

## Benchmarking the NV-DAC platform

To validate the NV-DAC system for characterizing high-pressure super-conductivity, we benchmark the method using the well-characterized cuprate superconductor Bi$_2$Sr$_2$CaCu$_2$O$_{8+\delta}$ (BSCCO)[63]. Our strategy combines conventional transport measurements with spatially resolved magnetic imaging using NV centres embedded directly in one of the culets of the DAC. Similar to the study of La$_3$Ni$_2$O$_7$ in the main text, this integrated approach allows us to precisely determine the super-conducting transition and directly image the associated Meissner effect, demonstrating the effectiveness of NV centres for probing superconductivity under extreme conditions.

The sample is mounted inside a DAC equipped with four electrical leads for transport measurements and an off-sample microwave antenna for NV-based magnetic field sensing, with rubies to calibrate pressure (Extended Data Fig. 2a). Transport measurements show a superconducting transition at approximately 92 K at 2.1 GPa (Extended Data Fig. 2d).

**Zero-field-cooled diamagnetism.** Following ZFC to a base temperature of 7.7 K, we apply a magnetic field of 158 G. We then record a wide-field map of $B/H$ by imaging the local magnetic responses from the shallow NV centres in the diamond culet beneath the BSCCO sample. Extended Data Fig. 2b shows representative raw spectra from a one-dimensional cross-section (indicated in Extended Data Fig. 2a), analogous to Fig. 1. Far away from the sample (at the top of the waterfall plot), the [111] NV resonances are split exactly by the expected value for $H = 158$ G. Approaching the sample edge, this splitting sharply increases, consistent with the compression of magnetic field lines near the boundary of a superconductor. Moving further into the sample interior, the resonances converge, indicating diamagnetic suppression. By fitting these resonances pixel by pixel, we obtain a 2D $B/H$ map illustrating BSCCO in the Meissner state (Extended Data Fig. 2c). The NV centres at the sample centre exhibit $B/H \approx 0$, confirming the expected bulk superconductivity. The enhancement due to field deformation at the sample edge is correspondingly strong.

**Field warming.** With the field $H = 158$ G fixed, we then warm the sample stepwise up to 95 K. Selected temperature-dependent $B/H$ maps are

shown in Extended Data Fig. 2c. The diamagnetic response gradually fades as temperature increases, with the Meissner effect disappearing around 90 K, consistent with the transport-derived $T_c$ of 92 K.

**Flux trapping.** Following another ZFC to 7.7 K, we apply a magnetic field $H = 200$ G (above the lower critical field of BSCCO at low temperature, $H_{c1} \approx 100$ G; ref. 64) and then quench the field. Imaging the remnant magnetic field $B$ (Extended Data Fig. 2e) shows that up to 60 G is trapped within the sample region, especially near the boundaries of the sample, with no field trapped outside the sample. This result confirms strong magnetic flux trapping in our pressurized BSCCO sample.

**Higher pressure.** Increasing the pressure to 7.7 GPa and performing a resistance measurement shows a slight reduction in $T_c$ to approximately 89 K (Extended Data Fig. 2d). At 7.7 K and an applied field of 158 G, we again observe diamagnetic suppression within the sample and magnetic field enhancement at its edges after ZFC. However, the overall response is weaker than at 2.1 GPa. Furthermore, the diamagnetic response exhibits a striped texture, possibly because of pressure-induced micro-fractures in the thin sample flake, which were not observable under optical microscopy, highlighting the unique diagnostic abilities of NV magnetometry.

These benchmarking experiments with BSCCO demonstrate that NV-DAC magnetometry reliably detects superconducting transitions and directly images superconducting diamagnetism under high-pressure conditions. Thus, this integrated method provides a robust and sensitive platform for exploring superconductivity in new materials under extreme conditions.

## Electronic transport for S1, S2 and S3

Electrical resistance was measured using the four-probe method with platinum leads pressed onto the sample (Extended Data Fig. 3c). In the main text, we show select transport measurements from S1, S2 and S3. Here, we present an expanded set of resistance curves. In Extended Data Fig. 3a, we list resistance curves taken over the full range of measured pressures of S1. The first signatures of a resistance drop appear at $\overline{\sigma_{ZZ}} = 16$ GPa. Extended Data Fig. 3b shows resistance measurements taken at $\overline{\sigma_{ZZ}} = 23$ GPa with different permutations of transport leads. The transport channel that probes the resistance of the sample between leads 2 and 3 shows essentially no superconducting transition. The region spanned by these leads is precisely the region of S1, which shows the least diamagnetism at this stress point (Figs. 1c and 2a). By contrast, the regions between leads 3 and 4, and likewise between 4 and 5, show more meaningful diamagnetism. Correspondingly, there is an appreciable transport signature in resistance measurements taken across these regions. This emphasizes the ability of our method to locally diagnose superconductivity in a multimodal fashion.

We show the transport for all pressures for S2 (Extended Data Fig. 3c) and S3 (Extended Data Fig. 3d). Although S2 shows superconducting behaviour starting at $\overline{\sigma_{ZZ}} = 13$ GPa, the transport of S3 shows no visible superconducting transition over the entire pressure range.

## Low-pressure ZFC-FW $B/H$ maps for S1, S2 and S3

ZFC-FW maps of diamagnetism in the main text were shown for pressures above the superconducting threshold for all samples. Here, we show low-pressure maps of $B/H$ for S1, S2 and S3 (Extended Data Fig. 4a–c). The maps are measured at temperatures both below and above the high-pressure $T_c$ of about 80 K. All high-temperature and low-pressure, $B/H$ maps show $B/H \approx 1$ over the sample region, until the normal stress exceeds 15 GPa and temperature drops below $T_c$ (for example, S1 $B/H$ map at 16 GPa and 35 K and S3 $B/H$ map at 20 GPa and 7.5 K), which is in agreement with the phase diagram shown in the main text. We also show the EDX mapping of S2 in Extended Data Fig. 4d.

## Extended temperature sweep ZFC-FW $B/H$ maps for S1

In the main text, we selected five temperature points to show the evolution of the magnetic response of S1 with temperature. Here, we focus on S1 at $\overline{\sigma_{ZZ}} = 21$ GPa and present a series of 18 maps of $B/H$ from 20 K to 101 K (Extended Data Fig. 5). All regions of non-trivial magnetic response $B/H \neq 1$ vanish above the temperature at which a kink appears in the electronic transport (Fig. 2c) at $T_c \approx 80$ K.

## Field cooling magnetic response for S1

The magnetic response maps shown in the main text were obtained in the ZFC-FW modality. We have also mapped $B/H$ while field cooling from $T > T_c$ at fixed $H = 150$ G at $\overline{\sigma_{ZZ}} = 23$ GPa in sample S1 (Extended Data Fig. 6a,c). The figure also shows a complete set of ZFC-FW data for comparison in the same stress environment (Extended Data Fig. 6b). Broadly, the most salient feature of this comparison is that the maps at the same temperature in the two protocols look similar, but the magnetic response is overall weaker in the field cooling measurement. That is, if a pixel shows a dia- or paramagnetic response that turns on below $T_c$ during the ZFC-FW protocol, it also does in the field cooling protocol with the same sign but typically smaller magnitude. This is consistent with the behaviour in dirty superconductors in which flux penetration in the mixed state becomes hysteretic because of pinning. We note that the electrical resistance curves measured during the two protocols are indistinguishable, suggesting that the extra flux penetrating the sample in the field cooling configuration is not significantly modifying the network carrying supercurrent.

## Flux trapping and analysis

The field-cooled flux-trapping protocol, which was previously described, was used for this experiment. Here, we note that the trapped flux that was frozen in across the superconducting transition during the field cooling procedure led to only a small signal. Challenges in directly measuring the flux included: (1) inhomogeneities in $E_0$ due to stress and electric-field disorder, which made the magnetic field contributions difficult to identify, and (2) in the small-field regime, the NV groups experience minimal $E_0$, leading to reduced distinguishability of all eight groups. As a result, the relevant peaks, which correspond to the NV$^{(0)}$ group, may be hard to fit. To rectify this issue, a differential approach was adopted, in which we compare ODMR spectra taken after the following steps (Extended Data Fig. 7a):

1. Perform ZFC to $T = 20$ K and $H = 0$ G and collect the ZFC spectrum.
2. Turn the field on and set $H = 150$ G.
3. Cycle back above $T_c$ to $T = 150$ K and perform field cooling back to $T = 20$ K.
4. Turn off the field and collect the field-cooled spectrum.

The first ZFC spectrum provides a reference measure of the local stress and electric field contributions to the peak splitting, and ultimately, we compute the trapped field by subtracting this in quadrature[18] from the final field-cooled spectrum:

$$\gamma B_Z^{\text{Trapped flux}} = \sqrt{\Delta \nu_{\text{FC}}^2 - \Delta \nu_{\text{ZFC}}^2} \tag{1}$$

This approach succeeds for 85% of the pixels in sample S1. There are two failure modes: first, some of the low-field spectra simply do not have identifiable peaks (13.3% of the pixels; Extended Data Fig. 7b, purple). Second, a small number of pixels (1.5%) have peaks that apparently shift inwards in the field-cooled final spectra by more than our noise floor of 5.6 MHz, because of electric field disorder in the absence of fields and stress; these pixels are marked in red. We display raw ODMR data comparing the ZFC and field-cooled spectra in Extended Data Fig. 7c.

In the main text, we discussed how regions of strong flux trapping in S1 correlated with regions of strong diamagnetism as observed in

ZFC-FW. At first, this seems counterintuitive: we expect flux to be preferentially trapped at crystalline defects, suggesting that regions with weak diamagnetism (that is, less pure or homogeneous superconductivity) should correlate with flux trapping. However, because $La_3Ni_2O_7$ superconductivity is already quite heterogeneous, we expect all parts of the crystal to have sufficient defects to trap flux. In this filamentary limit, we are more concerned with superconductivity being present to trap flux. So, we expect—and observe—that regions of greater diamagnetism trap more flux.

## Sum of flux

Magnetostatics dictates that the total magnetic flux through the entire sample chamber should remain the same, regardless of whether a sample is present or not. That is, $\sum_i B_i \approx NH$, where the index $i$ runs over all $N$ pixels in a spatial map, $B_i$ is the field measured at a given pixel and $H$ is the applied field. Equivalently, $\sum_i \frac{B_i}{NH} \approx 1$. To this end, we explicitly calculate the value of $\sum_i \frac{B_i}{NH}$ (across the entire sample chamber) for all three samples and show the values in Extended Data Table 1. For all temperatures and pressures, across all three samples, the value of $\sum_i \frac{B_i}{NH}$ is extremely close to unity. To ensure that this is not just a quantitative artefact, we also compute $\sum_i \frac{B_i}{NH}$ over just the superconducting regions and observe values consistent with an approximately 2–4% diamagnetic suppression in the superconducting region.

## High-pressure, optical and EDX experimental details

**Sample preparation.** Single-crystal sample growth is detailed in refs. 9,65. To prepare the sample for the high-pressure NV-DAC measurement, a piece of a $La_3Ni_2O_7$ single crystal was shattered by compression between two glass slides. Shards were subsequently chosen for prescreening by EDX, based on their size and shape relative to the DAC sample chamber. We then selected samples with varying degrees of stoichiometric inhomogeneity for the studies of S1, S2 and S3.

**Creation of NV centres.** For the high-pressure NV measurement, we use 16-sided standard design diamond anvils with [111]-crystal cut culets (Almax-easyLab). These anvils are polished from synthetic type Ib ([N] ≤ 200 ppm) single-crystal diamonds. The culet diameters are in the range of 300–400 μm. Similar to previous work, we perform $^{12}C^+$ ion implantation with multiple energy shots up to 450 keV with dosage up to $9.9 \times 10^{11}$ cm$^{-2}$ (CuttingEdge Ions) to generate a layer of vacancies of about 500 nm from the culet surface[21]. Following implantation, we perform vacuum annealing of the diamond anvils (at $P < 10^{-6}$ mbar) in a custom-built furnace at a temperature >850 °C for 12 h. During the annealing step, the vacancies become mobile and probabilistically form NV centres. The NV diamond is glued to the cylinder side of a miniature DAC machined out of beryllium copper (Cu–Be). A Pt foil with a width of 20 μm is incorporated near the culet to apply microwave radiation for spin-state control. We further use Pt foil to serve as the transport connection pads by compression against the sample crystal surface under pressure. On the opposing (piston) side, we glue a type Ia diamond anvil with [100]-crystal cut. A double-sided insulating gasket, fashioned out of rhenium (Re) foil and cubic boron nitride (cBN) mixed with epoxy, is used to isolate the transport leads and the microwave wire.

**Wide-field imaging in the AttoDRY800.** The NV magnetic and stress imaging of sample S1 and S2 are performed in a dry, closed-cycle cryostat (Attocube AttoDRY800) integrated with a custom-built wide-field fluorescence microscope. The continuous excitation beam from a 532-nm laser (Sprout Solo-10W) is diffused and refocused at the back of an objective lens (20× Mitutoyo Plan NA 0.42). The field of view is in the range of 200–300 μm. The fluorescence collected by the objective is separated from the excitation path by a dichroic mirror and collected by an electron multiplying charge-coupled device (CCD) (Princeton Instrument PROEM-HS: 512B eXcelon). A data

acquisition device (National Instruments USB-6343) is used for fluorescence counting as well as modulation of a microwave source (SRS SG384) for continuous wave ODMR measurements. The microwaves are amplified (Mini-Circuits ZHL-25W-63+) and subsequently channelled to the sample using coaxial cables and a custom-built circuit board. A vector external magnetic field is applied by running currents from a d.c. power supply (Keithley 2200-30-5 and Keysight e36154a) into custom-built electromagnet coils that sit outside the cryostat shroud. The $z'$-coil is circular and closer to the shroud. The maximum applied field is 240 G. The $x'$ and $y'$ coils are rectangular Helmholtz coils, which are further away from the shroud. Their maximum field is 50 G. A lock-in amplifier (Stanford Research Systems SR860) in combination with a voltage-controlled current source (Stanford Research Systems CS580) is used to measure the four-point resistance of the sample.

**Wide-field imaging in the AttoDRY2100.** NV magnetic and stress imaging of sample S3 was performed in a helium-gas-cooled closed-cycle cryostat (Attocube AttoDRY2100) integrated with a custom-built wide-field fluorescence microscope. A continuous 532 nm excitation beam (Gem SMD12-INV-v7) was spatially homogenized and focused through the back aperture of a long-working-distance objective (LT-APO/ULWD/VISIR, NA 0.35), yielding a field of view of 100–300 μm. Fluorescence emission was separated from the excitation path by a dichroic beam splitter and collected by an electron-multiplying CCD camera (Princeton Instruments ProEM-HS:512BX eXcelon). Signal acquisition and microwave modulation for continuous wave ODMR were performed using a National Instruments USB-6343 data acquisition card. Microwaves from an SRS SG384 source were amplified (Mini-Circuits ZHL-16W-43+) and delivered to the sample via coaxial cables and a custom-designed printed circuit board. A superconducting vector magnet integrated within the cryostat provided independently controlled magnetic fields up to 1 T along the $x'$-, $y'$- and $z'$-directions. Four-point resistance measurements were carried out with a Zurich Instruments MFLI lock-in amplifier, in combination with a Stanford Research Systems CS580 voltage-controlled current source.

**Stoichiometry measurements by EDX spectroscopy.** The EDX spectroscopy measurements were performed using a Gemini 360 field emission scanning electron microscope and an Oxford Instruments Ultim Max detector. The electron energy is fixed at 15 keV, and the beam current at 60 mA. The measured elements are calibrated by well-characterized standards $La_2O_3$ and Ni.

## Calibrating $\sigma_{ZZ}$ and ruby pressure

In the main text, we stress the importance of using $\overline{\sigma_{ZZ}}$ as a marker of sample stress, as opposed to ruby pressure. Here, we show a reasonably linear tracking of ruby pressure with $\overline{\sigma_{ZZ}}$ across all three samples (Extended Data Fig. 1c). This emphasizes our point that although ruby pressure is a reasonable proxy for the average stress conditions of the sample, it is imprecise in two key ways. First, ruby pressure is lower by a factor of about 0.73 compared with $\overline{\sigma_{ZZ}}$. This reflects the non-hydrosticity of the salt pressure medium; the in-plane stress is lower than the stress along the loading axis as we might expect for a solid. Second, ruby pressure fails to capture the several-GPa local variations of normal stress present across the samples.

## Calibrating magnetic fields

**Field calibration in the AttoDRY800.** For the S1 and S2 measurements performed in the Attocube AttoDRY800 cryostat, we use a set of electromagnets that are roughly pointed in orthogonal directions, denoted as $x'$, $y'$ and $z'$.

All frames are shown in Extended Data Fig. 1b and are discussed above in the section 'Notation'. The vector magnetic field **B** produced at the

sample is linearly related to the three currents $J_i'$ applied to each of the electromagnets,

$$B_i = C_{ij}' J_j' \qquad (2)$$

where we work in the NV[(0)] frame (repeated indices summed). To calibrate the matrix $C$, we perform a series of ODMR measurements on NV centres in the facet of the diamond, in which we expect an ambient pressure environment, to mitigate any stress background in the calibration. Furthermore, this is done in situ, such that, after calibration, the cell does not move within the cryostat and the calibration remains accurate within each experimental run.

We perform the calibration measurements by finding the current configurations that produce magnetic fields close to alignment along the four NV orientations. We start by roughly orienting the magnetic field along the [111] crystal direction of diamond (roughly along the DAC axis). By varying the applied currents $J_i'$, we adjust the applied field until the ODMR spectrum exhibits two widely spaced peaks because of the Zeeman splitting of the NV[(0)] group and two peaks at 1/3 the spacing of the outer pair. The degeneracy of the peaks of the other three groups corresponds to them all sharing an angle with cosine −1/3 with the magnetic field.

This current configuration defines $\mathbf{B} \propto \hat{z}$.

We then repeat the above alignment procedure for each of the remaining three NV groups $g = 1, 2, 3$ and thereby obtain reference current configurations that produce $\mathbf{B} \propto \hat{z}^{(g)}$.

At this point, we have an over-constrained set of linear equations to solve for the nine components of $C$. We use three of the measurements to solve for $\mathbf{C}$, and use the fourth one to estimate the error in the alignment:

$$\text{Error} = \frac{||\Delta v_{\text{calc}} - \Delta v_{\text{meas}}||}{||\Delta v_{\text{meas}}||} \qquad (3)$$

where $\Delta v$ is the Zeeman splitting of the widely spaced peaks from the fourth NV group. The typical error is below 1%, which corresponds to a misaligned angle below about 0.6°. We use the calibrated $C$ to determine the electromagnet current values required for any desired vector magnetic field $\mathbf{B}$.

**Field calibration in the AttoDRY2100.** Calibration of sample S3 in the Attocube AttoDRY2100 cryostat follows essentially the same procedure as for samples S1 and S2, but with additional refinement. As above, we seek the transformation matrix that enables calculation of fields in the NV frame. The AttoDRY2100 has three built-in orthogonal superconducting magnets, which simplifies our task. We perform the method described above to get an initial solution for $C$, and then refine these values with the following automated procedure. Thirty randomly oriented 100 G fields are applied to the sample using roughly aligned currents. For each, we obtain a corresponding ODMR spectrum in which the peaks are fitted using a peak finder that is seeded with the expected positions of the peaks from the initial calibration $C$ computed from the NV ground state Hamiltonian. We then use the measured peak locations to obtain a best-fit $\mathbf{B}$ from the model Hamiltonian.

We assemble the 30 applied current vectors $J_i'$ and best-fit measured field vectors $B_i$ into $30 \times 3$ matrices $A$ and $F$, respectively. We then compute the Moore–Penrose pseudo-inverse $A^+$ and form the transfer matrix

$$C_{ij'} = F_{i,m} A_{m,j'}^+ \qquad (4)$$

which formally gives the optimal solution to this over-constrained linear system (here, $m$ runs over the 30 measurements). This method allows us to refine our calibration with arbitrarily many test fields and can achieve misalignments <0.1°.

## Diamond anvil miscut

Polishing (111) cut anvils is technically challenging, and deviations between the culet plane and the ideal crystal plane up to $\theta < 5°$ are to be expected. That is to say, the [111] NV axis $\hat{z}$ has a miscut angle $\theta$ with respect to the culet normal $\hat{Z}$. This is shown in Extended Data Fig. 1a, in which we define a lab frame with $XYZ$ axes aligned so that $\hat{Z}$ is orthogonal to the culet and a rotated NV frame with $xyz$ axes aligned conventionally with respect to the [111] NV symmetry axis.

The magnetic field $\mathbf{B}$ and stress tensors $\overset{\leftrightarrow}{\sigma}$ extracted from NV splittings are most naturally constructed in the NV frame. Neglecting the miscut generally leads to negligibly small errors in $B_Z$ measurements and normal stress $\sigma_{ZZ}$ components, as these are only corrected at $O(\theta^2)$. However, it generally leads to more important corrections to the transverse components of these quantities. For example, the miscut rotates the large normal $ZZ$ stresses into the smaller shear components at $O(\theta^1)$. Thus, to correctly estimate the shear on the sample, given by the $\sigma_{*Z}$ components in the lab frame, we must take the miscut into account.

To rotate measured stress tensors into the lab frame from the NV frame we require the rotation matrix,

$$R_{lj} = [R_Z(\psi) R_Y(\theta) R_Z(\phi)]_{lj}$$

where the Euler angles $\phi$ and $\theta$ specify the orientation and magnitude of the miscut. The final rotation $\psi$ leaves the culet normal invariant and can be used to orient the lab $XY$-axes with the axes of the image plane in the CCD.

We measure the miscut angle with two independent methods. First, we perform single-crystal X-ray diffraction on the diamond and resolve the tilting of the diamond lattice with respect to the culet. Second, we load the DAC with $N_2$ pressure medium and pressurize to 9 GPa. The negligible ability of $N_2$ to sustain shear[43] means that $\sigma_{XZ}, \sigma_{YZ} \approx 0$ across the culet. We can, therefore, evaluate $\sigma^{\text{Lab}} = R\sigma^{\text{NV}}R^T$ and solve for the rotation matrix $R$ such that the $\sigma_{XZ}^{\text{Lab}} \approx \sigma_{YZ}^{\text{Lab}} \approx 0$.

The $N_2$ method gives a miscut magnitude of $\theta = 3.5(6)°$ and an azimuthal orientation of $\phi = -5(6)°$ for S1. XRD gives a miscut magnitude of $\theta = 3.02°$, but disambiguating the azimuthal angle proved difficult. We chose to use the miscut parameters derived from the $N_2$ method, noting that the XRD method is also more susceptible to small misalignments between the culet plane and the detector owing to imperfections in the XRD sample mounting. Regardless, we note that the two miscut angles agree within error, and these results are within the specifications of the diamond polishing method. We apply this rotation to measured stress tensors to extract the lab frame stress.

The effect of this rotation is rather small. As mentioned above, the adjustment is first order for shears, and results in a correction of about 1 GPa, and second order for normal stress, with a correction of about 0.1 GPa. We have checked that arbitrary (incorrect) miscut corrections with $\theta < 5°$ do not qualitatively change the shape of the phase diagrams and shear–superconductivity correlations presented in the main text, establishing that miscut corrections are minimal.

## ODMR peak fitting

A single wide-field image has $10^5$ pixels, each containing an ODMR spectrum. We use a highly parallelized fitting strategy to extract magnetic field and stress parameters. Each ODMR spectrum has up to eight NV resonances: a pair from each of the four NV groups. Rather than directly working with the two peak frequencies for each group $g$, it is more convenient to work with their average frequency $D_g$ and their half-difference $E_g$. For example, extracting $B_z$ requires finding the $E_0$ splitting, whereas a stress tensor solution generally necessitates finding the $D_g$ for all groups $g$. These tasks, naturally, require that we correctly identify which resonances come from which NV group, and fit their frequencies accurately. This is complicated by several factors: stress and field gradients in the sample broaden and weaken the resonances;

stress-induced symmetry breaking of the point group of the NV centre reduces peak height; and the laser and microwave polarizations are less effective at polarizing certain NV orientations. We use a custom-built MATLAB program to address these challenges. The operational procedures are as follows:

1. In each pixel, we deploy the standard MATLAB peak-finding routine to roughly identify the position of all NV resonances after first denoising the data with a Savitzky–Golay filter.

2. We then pair the ±1 resonances of a given NV group by their proximity to a regionally seeded value of either $D_g$ or $E_g$ using a maximum bipartite matching algorithm. As an example, let us focus on the [111] group. If a field of 100 G is applied along the [111] NV, we expect the resonances to have an $E_0 \approx 280$ MHz. Because $B$ varies at most by about 15%, and because the other NV groups will experience a splitting of about 1/3 of what is experienced by the [111] NV, the pair of peaks that has an $E$ splitting nearest to this seed is reliably the [111] NV. The limited variasion of $B$ makes $E_g$ seeding well suited for La$_3$Ni$_2$O$_7$ mapping; BSCCO (Extended Data Fig. 2) is better suited to $D_g$ seeding as in that sample $B$ varies greatly, whereas stress (which controls $D_g$) is relatively uniform.

3. We impose constraints on peak height, width, $D_g$ and/or $E_g$ to further enhance the accuracy in partnering resonances. Resonances from the same NV group are expected to have similar height and width by symmetry constraints.

4. In <1% of spectra within the sample chamber, this partnering fails because of the poor signal-to-noise ratio of one of the peaks in the pair. Thankfully, knowledge of either $D_g$ or $E_g$ allows the fit to proceed even if only one resonance is available. If we seek to fit $E_g$, we may extract $D_g$ from a dataset at a different temperature or magnetic field (which influences $D_g$ trivially), in which the formerly obscured peak is recovered. Likewise, fits of $D_g$ can use $E_g$ values obtained at a different temperature.

5. Once peaks are identified, resonance values are further refined by quadratic interpolation around the frequency points identified in the peak-finding step to mitigate sampling error (we sample our ODMR every 2 MHz).

6. The processes in steps 1–5 are repeated in parallel for all pixels, allowing for a mapping of $D_g$ or $E_g$ for the desired NV resonance.

Our automated method generally completes in <10 s.
From here, several further refinements are possible:

1. Our software contains a customized version of the QDMLab GPU fitting code developed in ref. 66. The method above identifies only the peak of the resonances, so for benchmarking, we use the QDMLab-based module for curve-fitting the whole resonances with Gaussian line-shapes, seeded by the results of the peak-finding procedure. In general, re-fitting with QDMLab does not significantly change the results of the peak-finding method for any of the data presented here.

2. In the case of high stress and low temperature, we often observe the inversion of contrast in the non-[111] resonances (see section 'Positive contrast'). In some high-stress datasets, we find several pixels in which these inverted resonances are the only ones present, and that both [111] resonances are weak. As we typically rely on the [111] resonances to determine $B_z$, this is a challenge. Knowing the projection of the magnetic field along any of the non-[111] NV subgroups is sufficient to extract $B_z$, given knowledge of the local orientation of the field. We can determine this orientation locally; adjacent pixels in which both [111] and non-[111] resonances are visible allow us to determine the relative projection of $B$ onto these groups. Generally, $B$ is essentially parallel to $H$ (which is along $\hat{z}$) owing to the rather small diamagnetism ($1 > B/H > 0.97$) in these regions.

3. Even after all these steps, we find that of order tens of pixels fail to automatically fit in typical magnetic $B/H$ maps. We manually fill in these rare pixels by assuming the $B_z$ is harmonic in the 2D plane and

visually checking that the ODMR spectrum has plausible, if weak, peaks at those positions.

4. For maps of stress, when the automated peak fitting procedure fails, we can often resolve peaks manually by reseeding the fitting procedure using rough peak positions determined by continuity with nearby pixels. Again, when we need to do manual reseeding, we confirm peak assignments visually within each ODMR spectrum.

## Positive contrast

Generally, NV resonances generate a fluorescence dip in ODMR spectra, referred to as negative contrast. However, we frequently observe inverted positive contrast peaks (for example, in Fig. 1e) at low temperature and high pressure ($\overline{\sigma_{ZZ}} \gtrsim 15$ GPa). We can follow the inversion of peaks as a function of decreasing temperature and increasing stress and field, and thereby identify them with their parent negative contrast resonances. At higher stress points, we make use of these resonances in extracting local magnetism and stress.

The phenomenon of positive contrast in pressurized NVs has been reported by multiple groups[21,60], without a comprehensive explanation for their origin. In a forthcoming paper[67], we provide a microscopic model for this intriguing and useful phenomenon.

## Constructing the local stress tensor from NV measurements

In the absence of any applied magnetic field, the average shift $D_g$ and splitting $E_g$ of the ODMR peaks of the NV centres in group $g$ are determined by the NV symmetry-preserving and symmetry-breaking stresses, respectively. With these eight quantities in hand, it is possible to reconstruct the local stress tensor $\overleftrightarrow{\sigma}$, whose six components are overconstrained. In practice, at zero magnetic field, the ODMR peaks because of the various groups $g$ overlap, and it is impossible to accurately resolve the peak positions and/or assign them to the correct groups.

The standard method for stress tensor extraction thus makes use of two kinds of ODMR measurements at a given point[18], both of which exploit carefully aligned applied magnetic fields. First, the 'D' measurements are performed by applying a longitudinal magnetic field along the NV$^{(g)}$ axis to split the resonances of group $g$ from those of the other three groups.

Second, the 'E' measurements require a magnetic field that is perpendicular to the relevant NV$^{(g)}$ axis. This now splits away the three other groups and allows resolution of the two central peaks of interest.

However, there are confounding effects. To understand these, we recapitulate the effective stress Hamiltonian governing the NV spin sub-levels, written in the local NV frame[18,60,61]:

$$H_\sigma = \Pi_z S_z^2 + \Pi_x(S_y^2 - S_x^2) + \Pi_y\{S_x, S_y\} \tag{5}$$

where

$$\Pi_x = -(b+c)(\sigma_{yy} - \sigma_{xx}) + \left(\sqrt{2}b - \frac{c}{\sqrt{2}}\right)(2\sigma_{xz}) \tag{6}$$

$$\Pi_y = -(b+c)(2\sigma_{xy}) + \left(\sqrt{2}b - \frac{c}{\sqrt{2}}\right)(2\sigma_{yz}) \tag{7}$$

$$\Pi_z = (a_1 - a_2)(\sigma_{xx} + \sigma_{yy}) + (a_1 + 2a_2)\sigma_{zz} \tag{8}$$

and $\{a_1, a_2, b, c\}$ are the stress susceptibilities of the NV centre[18,62]. The $\Pi_z$ term shifts the two ODMR peaks corresponding to the $0 \leftrightarrow \pm1$ transitions together, directly contributing to $D$, whereas the $\Pi_x$, $\Pi_y$ parts of the stress couple $-1 \leftrightarrow +1$ and contribute to the relative splitting $E$.

Extracting the stress contribution $\Pi_z$ from the 'D' measurement is relatively straightforward, as $D = \Pi_z + D_{gs}(T) + \frac{(\gamma B_\perp)^2}{D_{gs}(T)}$, including both zero-field splitting $D_{gs}$ and leading order magnetic effects. The

temperature-dependent zero-field splitting $D_{gs}(T)$ is well characterized[68] and easy to subtract. Moreover, as it is about 3 GHz, the contribution from transverse magnetic fields $\gamma B_\perp < 3$ MHz (which are generated from field misalignment) is negligible. As such, $\Pi_z$ is robust against magnetic fields and may be reliably extracted.

By contrast, the 'E' measurements are much more challenging. First, the sizable transverse magnetic fields (about 100 G) required to split away the non-group $g$ peaks also tend to mix the NV spin sub-levels, which reduces the ODMR contrast. This is compounded by a similar reduction of contrast because of NV symmetry-breaking stress, which is non-trivial in the pressure range of this work. Even when the group $g$ peaks can be resolved, $E$ is a complicated function of $\Pi_x$, $\Pi_y$, **B** and contributions due to inhomogeneous internal electric fields that produce an ambient (no stress) effective splitting $\mathcal{E}_0$ (ref. 18). The $\mathcal{E}_0$ contribution combines with stress ($\Pi_x$, $\Pi_y$) by an elliptic integral; this composite quantity combines in quadrature with $B_z$ and trigonometrically with ($B_x$, $B_y$). Extracting the stress contributions from the $E$ measurement is highly nonlinear and is very sensitive to slight misalignments in the applied magnetic field. Finally, we note that there is further experimental uncertainty in the stress susceptibilities ($b$, $c$) by which ($\Pi_x$, $\Pi_y$) couple because of the aforementioned confounding effects of electric and magnetic fields (which demonstrably affect their calibration[18]).

In short, although wide-field $D_g$ maps for each of the four NV groups can be extracted reliably, the $E_g$ maps are generally much more challenging because of reduced ODMR contrast, tight magnetic field orientation tolerances and uncertainties in the relevant susceptibilities. We can take advantage of two simplifications even in the absence of $E_g$ data. First, only three components of the stress tensor are physically relevant to the sample—the traction $\sigma_{*z}$—and we need not reconstruct the full stress tensor. Second, the stress field must satisfy the constraints of mechanical equilibrium. In particular, the stress at the surface of the diamond anvil is dominantly $\sigma_{ZZ}$ and even in the presence of local surface shears, we expect the internal shear components of the stress to be very small. Building on this assumption, we define a method to reconstruct the normal stress $\sigma_{ZZ}$ and shear $\tau$ from the four $D_g$ values alone.

To test the approximate 4D method, we benchmark it against full reconstruction using a standard 4D + 2E method at a collection of sampled points at which the relevant E peaks were readily resolved. We find that the two methods produce linearly correlated results for the shear traction $\tau$, with a slope that characterizes the error of the accepted assumptions, which we use for rescaling all shear stresses obtained by the 4D method.

**The 4D method.** The 4D method is motivated by the observation that the stress field throughout a linear elastic medium, such as the diamond anvil, is determined by the three components of the surface traction $\tau$, $\sigma_{ZZ}$. Finite-element simulations of diamond suggest that the components $\sigma_{XX} - \sigma_{YY}$ and $\sigma_{XY}$ generated by these tractions are small. Thus, we make the simplifying assumption that these two components of the stress tensor vanish—and thus the four measured $D_g$ values—are sufficient to fix the four remaining components of $\overset{\leftrightarrow}{\sigma}$.

To recombine the four measured $D_g$ values into an estimate of the stress tensor in the lab frame, it is convenient to re-express the NV stress susceptibility in terms of tensorial quantities:

$$D_g = Y^{(g)}_{IJ}\sigma_{IJ} \tag{9}$$

$$Y^{(g)}_{IJ} = 3a_2\hat{z}^{(g)}_I\hat{z}^{(g)}_J + (a_1 - a_2)\delta_{IJ} \tag{10}$$

Here, the $D_g$ susceptibility $Y^{(g)}$ is a symmetric rank-2 tensor, which is invariant under the point group of NV group $g$ and $\hat{z}^{(g)}$ is the $z$-axis of group $g$. We take the susceptibilities $\{a_1, a_2\} = 2\pi \times \{4.86(2), -3.7(2)\}$

MHz GPa (ref. 69). Given the four measured values of $D_g$ at a point, and the vanishing internal shear assumption, it is straightforward to solve for the stress tensor $\sigma_{IJ}$ in the lab frame.

We note that several physically important components of the stress tensor are completely determined even without the assumption of vanishing internal shear. These include the pressure $p = \frac{1}{3}\sigma_{ii}$ and the normal stress in any of the NV-axis directions, notably $\sigma_{ZZ}$, which, up to a small miscut correction, is the normal component of the stress on the sample chamber. In particular, noting that $\sum_g \hat{z}^{(g)}\hat{z}^{(g)} = 4\mathbb{I}/3$, we obtain $p = \sum D_g/(12a_1)$.

**Benchmarking with the 4D+2E method.** We validate the 4D method by comparing it with the standard stress tensor extraction strategy, using, apart from the 4D values, 2E values to fully constrain the last two components of $\overset{\leftrightarrow}{\sigma}$. We note that the normal component of the boundary traction $\sigma_{ZZ}$ is the same in both methods (up to small miscut corrections). In Extended Data Fig. 8a, we randomly sample points on S1 at which the ODMR spectrum for the E measurements showed clearly resolvable peaks and calculate the shear traction $\tau$ using both methods. We observe a strong linear correlation, $\tau_{4D} \approx 1.8\tau_{4D+2E}$ (Extended Data Fig. 8b). The fact that the slope is not one and that there is some offset may be attributed to both uncertainties in the standard method (several GPa) and uncertainties in the susceptibilities, especially $b$, $c$ (ref. 18). We consider $\tau_{4D+2E}$ the true shear stress and rescale all $\tau$ obtained by the 4D method by this factor of 1.8.

The shear traction $\tau$ in the 4D method is independent of $a_1$ and thus depends inversely on the susceptibility $a_2$. Changing this susceptibility uniformly rescales the traction components. Various groups have reported different values of $a_2$, such as $a_2 = -6.5$ MHz GPa$^{-1}$ (ref. 70) and $a_2 = -2.51$ MHz GPa$^{-1}$ (ref. 61).

We use the susceptibilities calculated in ref. 69. Future work is needed to clarify the true value of this susceptibility, but adjusting our results will simply require the multiplication of some constant to all shear values in our work (and a similarly simple correction to $\sigma_{ZZ}$) and not qualitatively change the shape of the phase diagrams we extract.

**Extraction of normal component.** For low-stress datasets in S1 ($\overline{\sigma_{ZZ}} < 17$ GPa), we did not take separate measurements of $D_{1-3}$. However, when $H$ is applied along $\hat{z}$, we see that the non-[111] resonances are strongly overlapping, as they are degenerate under this magnetic field. This indicates that $D_1$, $D_2$ and $D_3$ are approximately equal and given by the average of the two visible peaks. This permits the construction of $\sigma_{ZZ}$.

## Pixel registration

As pressure varies in a DAC, the sample may slightly change shape and/or position because of the pressure gradient. Furthermore, temperature cycling can lead to subtle translations in the sample due to piezo drift. Thus, it is crucial to register the multimodal information to a common set of pixel coordinates.

Here, we show how to register maps from different pressure points and from different probes.

To register between different probes at the same pressure (NV fluorescence, EDX and white light), we manually select key features such as the ruby, edge of the sample and the leads as control points (no fewer than 10 features) and use these to calculate best-fit projective transformations between the images.

To register between different pressure points, we rely on white-light images, as these have the most visible structure to correlate. In this case, we select invariant key features using the Speeded-Up Robust Features algorithm in MATLAB. We then use these features as control points to calculate the best-fit affine transformation between different pressure points. We restrict to affine transformations, rather than the more general projective transformations permitted between probes, as the white-light imaging optics are fixed across all pressures.

### Constructing the pixel-based phase diagram

In the main text, we present phase diagrams in $\sigma_{zz}$–$\tau$–T space created from aggregated NV data across all samples. At a high level, we begin by selecting all the pixels that fall within a defined sample-spanning 'region of interest' across stress and temperature for each sample. Then, we define a vector of stresses, temperature and $B/H$ for each pixel by the collation of the aligned magnetic and stress maps. Finally, we plot these points in the respective coordinate spaces to generate the phase diagrams.

Now, we will detail the specific procedures for making the two 2D phase diagrams and the full 3D phase diagram, as well as the stoichiometric phase diagram.

**$\sigma_{zz}$ compared with $T$.** For S1, S2 and S3, maps of $\sigma_{zz}$ and $B/H$ are constructed at all stress and temperature points.

For each sample, these maps are aligned with respect to each other, as previously discussed. The goal of this phase diagram is to depict the effect of $\sigma_{zz}$ variation on $B/H$, so we mask out those parts of each sample that never conclusively superconduct at any point in our dataset. The samples contain substantial regions that never superconduct, for reasons that are evidently beyond the scope of stress (for example, chemical inhomogeneity in S3 and possible oxygen inhomogeneity or layer stacking disorder in S2). We could, in principle, generate this phase diagram by collecting pixels from the whole sample region, but in practice this generates a qualitatively identical phase diagram with a worse signal-to-noise ratio because of the integration over tens of thousands of 'dead' pixels. Similarly, we mask out sample regions with high shear (above about 2 GPa).

Doing this across all datasets yields pixels of the order of one million. We plot these points in ($\sigma_{zz}$, $T$) space with their colour controlled by $B/H$ and find excellent coverage of phase space—emphasizing how inhomogeneity can be leveraged as a metrological resource. For denoising purposes, we perform $k$-means clustering of these points ($k = 250$) after first temporarily rescaling the temperature axis (by a factor of 1/20, so as to roughly equate the typical spacing of points along both axes before clustering). We set a superconductivity threshold, $B/H = 0.98$, which used to define the diamagnetic suppression, below which we consider a local region to exhibit superconductivity; specifically, we use a sigmoidal low-passer ($k = 20$) centred around this value. We note that we set $B/H$ to unity for all regions with $B/H > 1$; these regions are non-superconducting and correspond, for example, to regions of magnetic-field enhancement. Using these clustered points, we then interpolate along both axes to fill the phase diagram. We emphasize that our results are relatively insensitive to choices of $k$ or axis rescaling.

**$\sigma_{zz}$ compared with $\tau$.** Similarly, we generate vectors from our maps of $\sigma_{zz}$, $\tau$ and $B/H$. We restrict ourselves to data with $T \leq 35$ K: as in the previous section, the diagram that integrates over all temperatures is qualitatively identical, but with a worse signal-to-noise ratio. We generally lack shear data for low-stress datasets ($\overline{\sigma_{zz}}$ less than 16 GPa, 10 GPa and 3 GPa for S1, S2 and S3, respectively), but these datasets have very weak to no magnetic signatures, so in practice their absence does not influence the constructed superconducting dome. We perform $k$-means clustering ($k = 85$) and use the same sigmoidal low-passer, with no axis rescaling necessary. We linearly interpolate to fill the phase diagram.

**Three-dimensional $\sigma_{zz}$–$\tau$–$T$ phase diagram.** In this plot, we eschew $B/H$ colouration for visual simplicity and instead construct a phase boundary by plotting $T_c$ for each ($\sigma_{zz}$, $\tau$) coordinate. To do this, we take the list of pixels from the previous step and bin them into ($\sigma_{zz}$, $\tau$) cells of side length 2 GPa by 1 GPa. For each cell, we fit a sigmoidal function $B/H = a + (1 - a)\left(1 - \frac{1}{1 + \exp(k(T - T_{mid}))}\right)$ to extract the superconducting-normal crossover $T_c = T_{mid} + 1/k$. In the main text, we define $T_c$ as the onset of superconductivity in transport and diamagnetism, which we

model here as one characteristic sigmoid width ($1/k$) above the crossover temperature. These $T_c$ values are splined together along both axes, and then the spline curves are linearly interpolated to generate the full surface. We note that this interpolation is done down to 0 on all axes, which requires extrapolation. However, we emphasize that this 3D phase diagram is in excellent accordance with the two 2D phase diagrams, although it was generated in a different way.

**La:Ni compared with $T$.** In Fig. 5c, we use EDX to determine the correlation of the La:Ni ratio with diamagnetism. This was done by binning pixels by stoichiometry intervals 0.2 wide, and calculating the average $B/H$ of the bin. Here, we extend this analysis across the temperature axis; we plot a 2D La:Ni compared with $T$ phase diagram in Extended Data Fig. 5e. Similar to the other 2D phase diagrams, we linearly interpolate the binned points in this phase space. We use a sigmoidal low-passer threshold of $B/H \approx 0.996$, reflecting the broadly weaker diamagnetism of S3. As expected, this phase diagram reproduces the two qualitative features observed in the main text: (1) $T_c \approx 60$ K and (2) an optimal La/Ni ratio of about 1.5, but also provides more quantitative information in the temperature–stoichiometry plane.

### Estimating superconducting volume fraction from stray field measurements

Our local measurements of $B/H$ provide an alternate route to extract the superconducting volume fraction of the sample.

Our NV centres (embedded in the anvil culet) are about 500 nm from the sample, which is significantly smaller than the sample dimensions (about 10–100 μm). Thus, we treat the NVs as measuring the stray field at the surface of the sample. To extract the superconducting volume fraction $f$ from our experimentally measured average diamagnetic suppression, $\langle B/H \rangle$, we work with the following simple model. Assume that a fraction $f$ of the sample is a perfect superconductor with $\chi_{SC} = -1$, and the remaining fraction of the material has a paramagnetic response, that is, $\chi_{PM} \approx 0$. The effective magnetic susceptibility of the sample is then given by

$$\chi_{eff} = f\chi_{SC} + (1 - f)\chi_{PM} \approx -f \qquad (11)$$

As our NVs effectively measure the stray field at the surface of the sample, we can approximate $\chi_{eff} \approx \langle B/H \rangle - 1$ and therefore $|\chi_{eff}| \approx 1 - \langle B/H \rangle$. Thus, for our three samples, we obtain a superconducting volume fraction of 3%, 0.3% and 0.5% for S1, S2 and S3, respectively. We may compare these volume fractions to those extracted in the literature using conventional methods involving a.c. or d.c. susceptibility measurements. Broadly, the highest volume fractions (>40%) are seen for the most hydrostatic pressure media, such as helium[9]. This aligns with our finding that shear quenches superconductivity; more hydrostatic pressure media are less able to support shear.

### Wohlleben effect

S1 exhibits intriguing magnetic behaviour: strong regions of paramagnetism that seem spatially separated from superconducting diamagnetism, but nevertheless vanish simultaneously with it above $T_c$. Although we expect, as in BSCCO, to see a halo of $B/H > 1$ surrounding diamagnetic regions (Extended Data Fig. 2c) as a result of field expulsion, the paramagnetism in S1 is unusual in its asymmetry (appearing predominantly on the left edge of the sample) and in its relative disconnection from the superconducting regions. In the main text, we connect this behaviour to the Wohlleben effect, which we will explore further here. This effect has been extensively observed and documented in granular superconductors[46,48]. For example, some models rely on Josephson coupling between discrete superconducting grains, leading to spontaneous currents, which generate the paramagnetism[71]. The fact that granularity seems to be crucial for observing this type of paramagnetic enhancement seems to point in the right direction for our La$_3$Ni$_2$O$_7$ samples, which exhibit strong heterogeneity at the μm scale.

Finally, although the Wohlleben effect has most often been examined under field-cooled protocols in previous works, some studies report a small paramagnetic signal in zero-field-cooled measurements a few kelvin below $T_c$ (ref. 72). By contrast, we observe a clear ZFC paramagnetic response across the entire sub-$T_c$ range, with a magnitude that is approximately proportional to the diamagnetic signal.

At a high level, some version of the Wohlleben effect seems like the most reasonable explanation for the data we see, given its robustness across pressure points and samples and its appearance in our granular $La_3Ni_2O_7$ system and not in other clean systems that we have studied such as BSCCO.

## Gradients of stress

Another way hydrostaticity may be violated is through stress gradients. Here, we ascertain if these stress gradients correlate with diamagnetic textures. In Extended Data Fig. 9a, we reproduce the diamagnetic map of S1 at $\overline{\sigma_{ZZ}}$ = 21 GPa and 20 K. In Extended Data Fig. 9b,c, we plot the gradients in $\sigma_{ZZ}$ and $\tau$, respectively. We then draw a white oval around the regions of maximal $\sigma_{ZZ}$ and a yellow oval around the regions of $\tau$, and also plot these ovals on Extended Data Fig. 9a. We observe that large gradients in either $\sigma_{ZZ}$ or $\tau$ do not suppress superconductivity. So, it seems that stress gradients do not correlate with the absence of superconductivity.

## Data availability

Published data are available at https://zenodo.org/records/17971352.

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

**Acknowledgements** We thank M. Wang and M. Huo for providing the nickelate samples used in this study. We also acknowledge G. Gu for supplying the BSCCO samples used for benchmarking. We thank the Fischer lab for generous access to their high-pressure sample preparation equipment. We thank T. Cavanaugh, S. Kraemer and S.-L. Zheng for their assistance in using shared facilities. These include the Harvard University Center for Nanoscale Systems (CNS), a member of the National Nanotechnology Coordinated Infrastructure Network (NNCI), supported by the National Science Foundation under NSF award no. ECCS-2025158 and the X-Ray Core facility, supported by the Major Research Instrumentation (MRI) NSF award no. 2216066. This work is supported by the Brown Institute for Basic Sciences. S.V.M. acknowledges support from the National Science Foundation Graduate Research Fellowship under grant no. DGE-1752814. V.I.L. acknowledges support of the NSF (DMR-2246991 and CMMI-2519764), ARO (W911NF2420145) and Iowa State University (Murray Harpole Chair in Engineering). C.R.L. is grateful for support through a Martin Gutzwiller Fellowship at MPIPKS, where his work was in part supported by the Deutsche Forschungsgemeinschaft under grant no. SFB 1143 (project ID 247310070) and the cluster of excellence ct.qmat (EXC 2147, project ID 390858490). N.Y.Y. acknowledges support from a Simons Investigator award.

**Author contributions** S.V.M., E.W., Z.W. and B.C. prepared samples, performed experiments and data analysis. S.V.M., Z.W., A.G. and B.C. developed the analysis software. S.V.M., Z.W. and N.C.J. worked on theoretical models and simulated the effect of stress on the NV centre, with guidance from V.I.L., C.R.L. and N.Y.Y.; S.V.M., E.W., Z.W. and E.A.R. calibrated the diamond miscut. C.R.L. and N.Y.Y. proposed and interpreted the investigations of the sample, and supervised the project. S.V.M., E.W., Z.W., B.C., N.C.J., C.R.L. and N.Y.Y. wrote the paper with input from all authors.

**Competing interests** Harvard University (co-inventors S.V.M., E.W., Z.W., B.C., N.C.J., C.R.L. and N.Y.Y.) filed for a provisional patent that relates to wide-field imaging of the stress tensor under pressure using spin defects embedded in a DAC.

**Additional information**
**Correspondence and requests for materials** should be addressed to C. R. Laumann or N. Y. Yao.

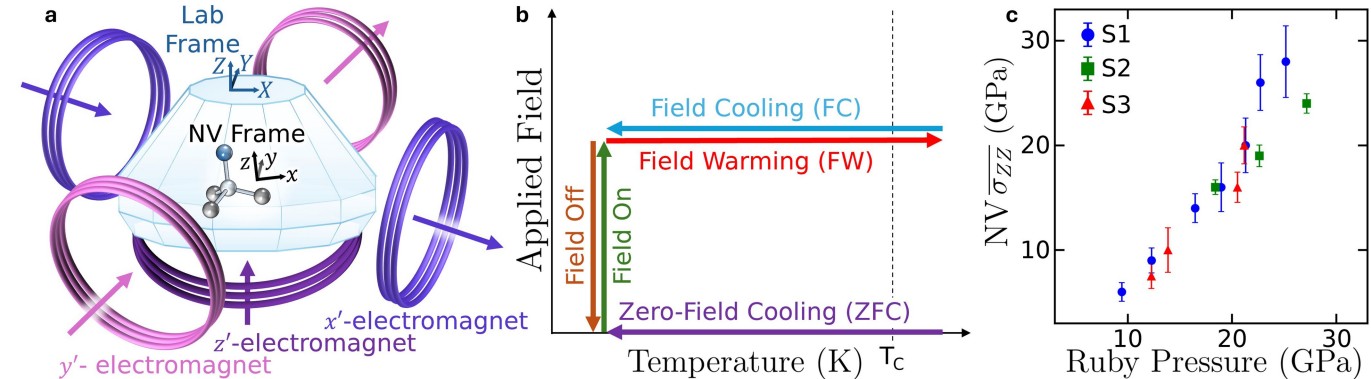

**Extended Data Fig. 1 | Schematic of experimental protocols. a**, The lab frame is defined by the culet normal $Z$ and the $X$, $Y$ axes of the CCD image plane at the culet. The global NV frame (that of the [111] NV$^{(0)}$ group) has $z$ aligned with the lab frame up to the miscut of the (111)-polished culet, and $x$ aligned with a neighboring carbon bond. The electromagnets define an additional $x'$, $y'$, $z'$ set of directions. **b**, Various steps in the measurement protocols with their corresponding names and abbreviations. **c**, Average NV normal stress $\overline{\sigma_{ZZ}}$ versus calibrated ruby pressure for samples S1, S2, and S3.

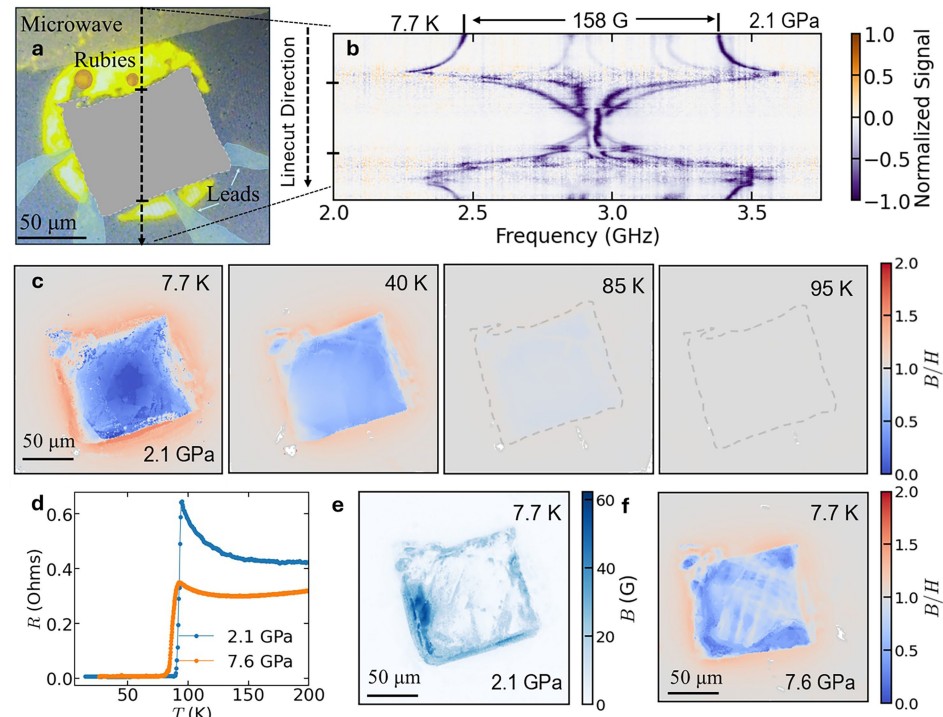

**Extended Data Fig. 2 | Benchmarking the widefield NV-DAC system with BSSCO. a**, Optical image of the Bi$_2$Sr$_2$CaCu$_2$O$_{8+\delta}$ sample within the diamond anvil cell. Rubies (pink) serve as pressure calibrants, four electrical leads (light blue) enable transport measurements, and a microwave antenna (light yellow) facilitates NV-based magnetic field sensing. The dashed line indicates the linecut used in (b). **b**, Raw ODMR spectrometry data recorded at 7.7 K under an applied field of 158 G. The color scale indicates the normalized NV fluorescence intensity, with darker regions corresponding to the resonance dips. **c**, Spatial mapping of $B/H$, measured at 2.1 GPa across various temperatures under a

$H$ = 158 G field, where $B$ is the magnetic field detected by the NV centers and $H$ is the applied field. **d**, Resistivity versus temperature curves measured at 2.1 GPa (blue) and 7.6 GPa (orange), indicating a shift in the superconducting transition from approximately 92 K to 89 K with increasing pressure. **e**, Map of the remnant magnetic field at 2.1 GPa and 7.7 K after zero-field cooling to 7.7 K, applying a 200 G field at the base temperature (above $H_{c1}$), and then quenching the field to 0 G, revealing trapped magnetic flux. **f**, Spatial mapping of $B/H$ at 7.7 GPa and 7.7 K following zero-field cooling and the application of a 158 G field.

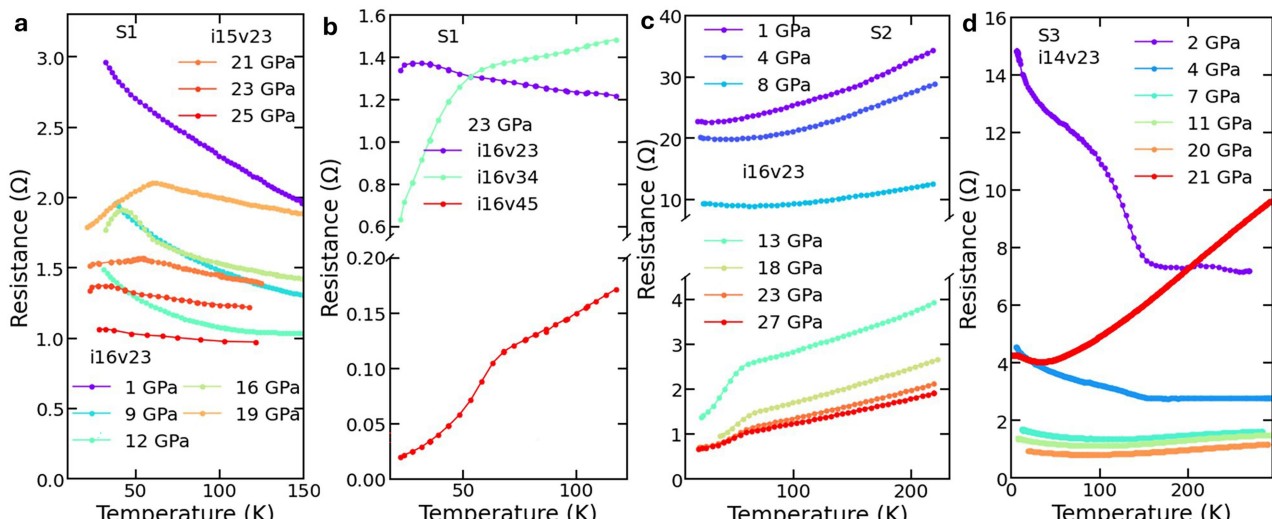

**Extended Data Fig. 3 | Temperature-dependent resistance $R(T)$ of samples S1-S3 under varying applied normal stresses $\overline{\sigma_{ZZ}}$. a**, $R(T)$ for S1 at $\overline{\sigma_{ZZ}}$ = 1, 9, 12, 16 and 19 GPa, (measured with current leads 1 and 6, voltage leads 2 and 3) and at $\overline{\sigma_{ZZ}}$ = 21, 23 and 25 GPa, (measured with current leads 1 and 5, voltage leads 2 and 3) respectively. The first clear drop in the resistance of S1 appears at $\overline{\sigma_{ZZ}}$ = 16 GPa. Lead numbers are in Fig. 1(c). **b**, $R(T)$ of sample S1 at $\overline{\sigma_{ZZ}}$ = 23 GPa, measured with current applied between leads 1 and 6 and voltage measured between leads 2 and 3 (purple curve), 3 and 4 (green), and 4 and 5 (red), respectively. Lead numbers are in in Fig. 1(c). **c**, Resistance of sample S2 as a function of temperature at $\overline{\sigma_{ZZ}}$ = 1, 4, 8, 13, 18, 23 and 27 GPa, measured with current passing through leads 1 and 6 and voltage measured between leads 2 and 3. A resistance drop emerges in S2 at $\overline{\sigma_{ZZ}}$ = 13 GPa. Lead numbers are in the inset to Fig. 3(c). **d**, $R(T)$ of sample S3 at $\overline{\sigma_{ZZ}}$ = 2, 4, 7, 11, 20 and 21 GPa, measured with current leads 1 and 4 and voltage leads 2 and 3. No superconducting transition observed up to $\overline{\sigma_{ZZ}}$ = 21 GPa. Lead numbers are in Fig. 5(a).

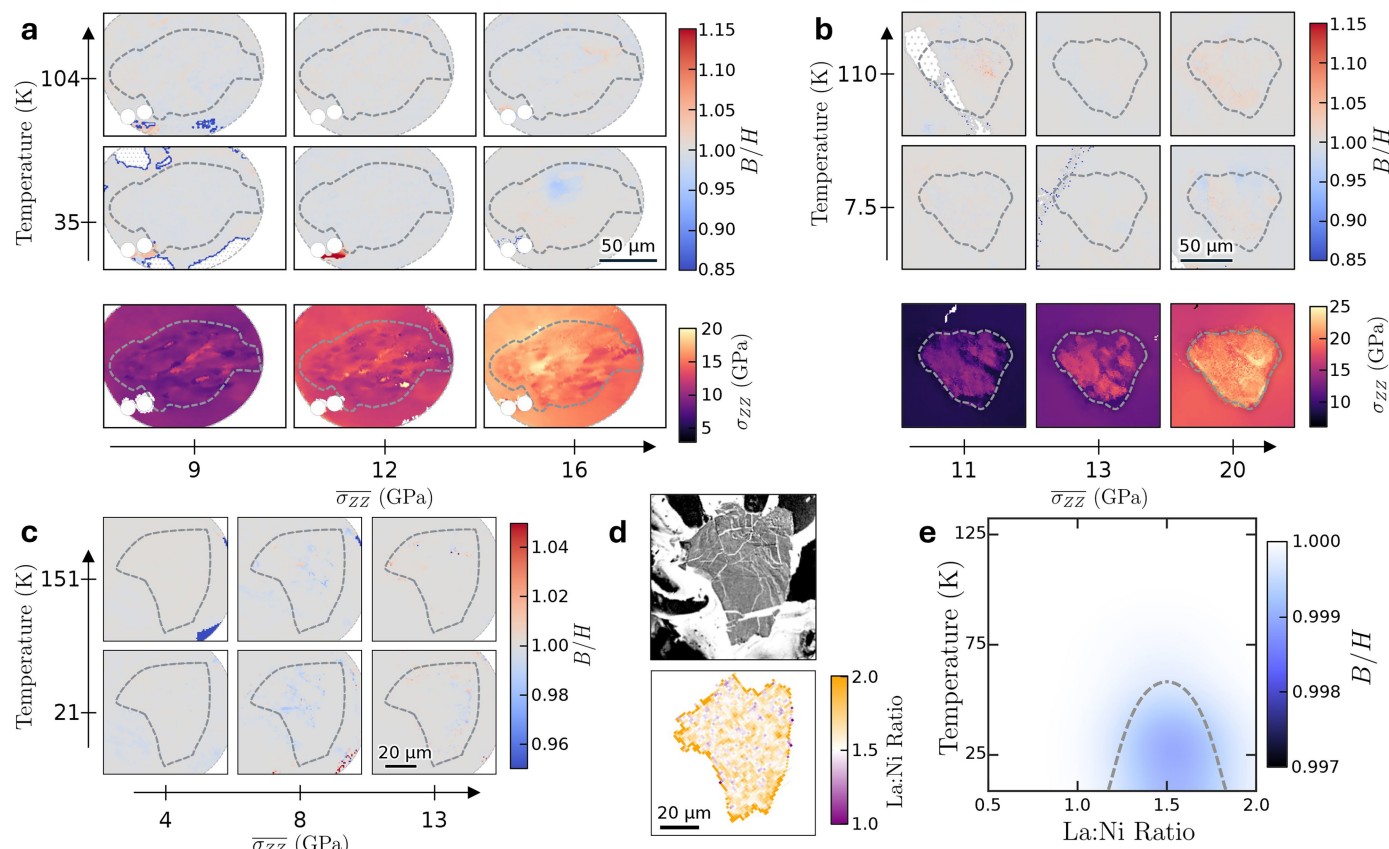

**Extended Data Fig. 4 | Low pressure ZFC-FW *B/H* maps for all samples.**
**a**,**b**, Sample S1 and S3's sub-micron resolution maps of the ZFC-FW magnetic ratio *B/H* and corresponding $\sigma_{zz}$ maps for $\overline{\sigma_{ZZ}}$ lower than 20 GPa. No evidence of superconductivity is visible until $\overline{\sigma_{ZZ}}$ > 15 GPa and *T* < 40 K. **c**, Sample S2's sub-micron resolution maps of the magnetic ratio B/H for $\overline{\sigma_{ZZ}}$ lower than 18 GPa.

None of the magnetic maps show conclusive evidence for superconductivity. **d**, (above) SEM image of S2 after recovery from the high pressure cell. (below) Spatial map of the La:Ni ratio from EDX showing many regions with La-rich composition. **e**, Superconducting dome in the La:Ni ratio–temperature plane. The dome is centered around the ideal La:Ni ratio of 3:2.

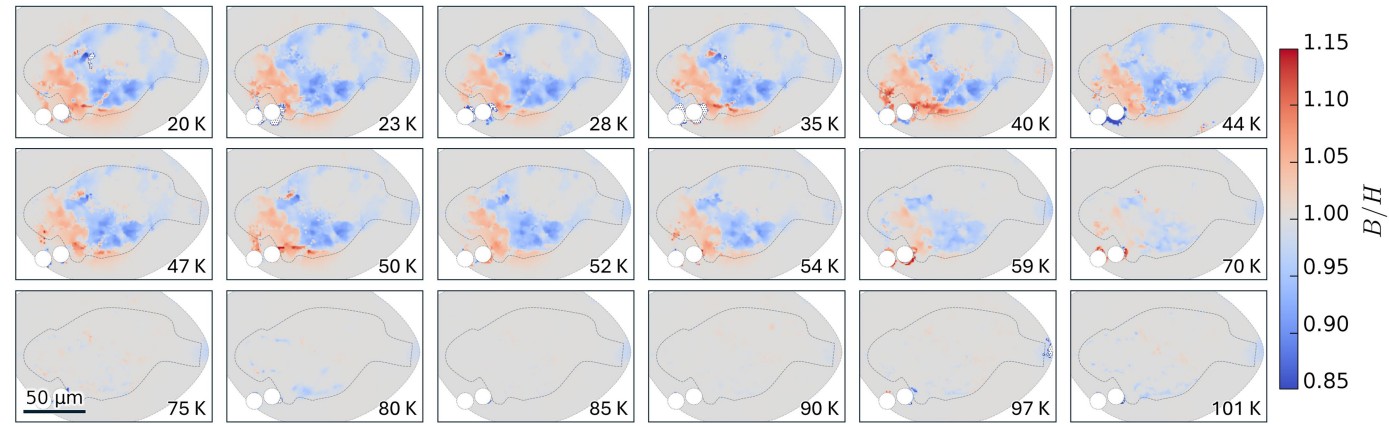

**Extended Data Fig. 5 | Dense temperature sweep of S1 ZFC-FW $B/H$ response at $\overline{\sigma_{zz}}$ = 21 GPa.** The temperature ranges from 20 K to 101 K in ~5 K steps.

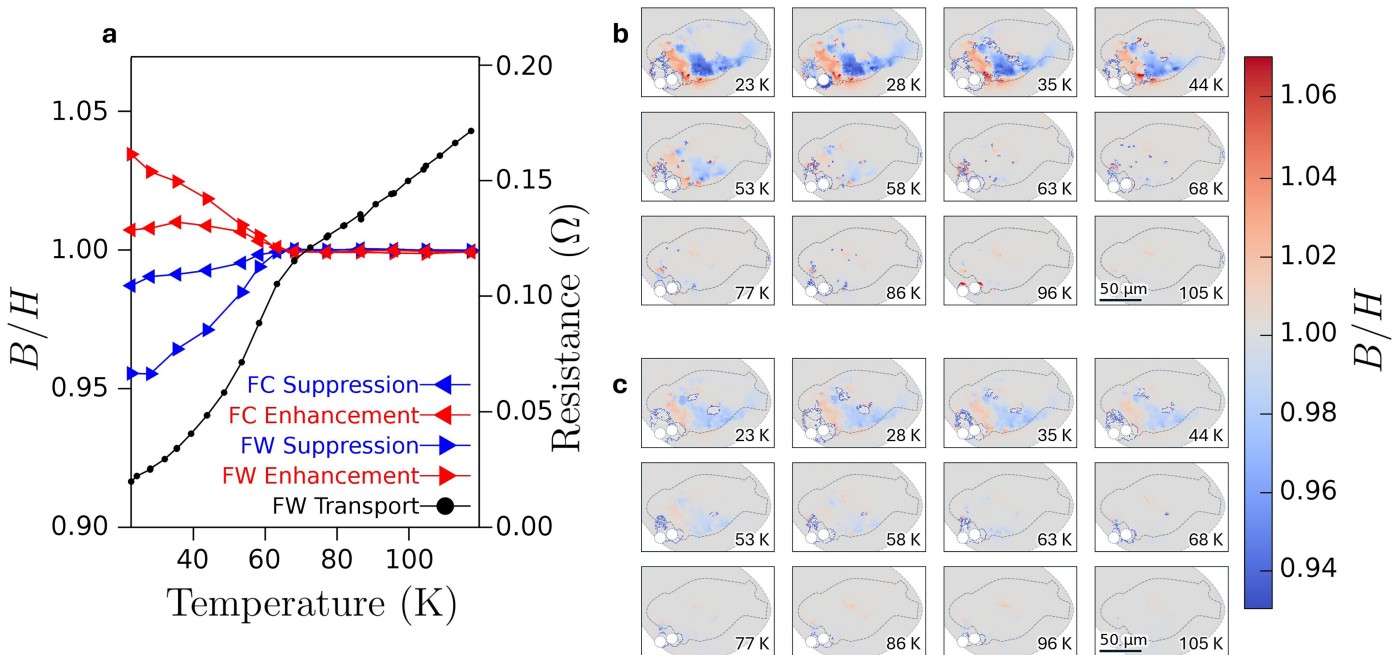

**Extended Data Fig. 6 | Field cooling study of magnetic response in S1 at $\overline{\sigma_{zz}}$ = 23 GPa. a**, The resistance measured in sample S1 shows no significant differences between FC and ZFC-FW protocols. Here, we show the ZFC-FW resistance. The local $B/H$ at different spatial points begins to deviate from 1 at the same $T_c$ in both conditions, but is quantitatively smaller in amplitude under FC conditions. The red (blue) curves correspond to the point at the pink (purple) star in Fig. 2(a). **b**, ZFC-FW $B/H$ maps. A subset of these panels are shown in Fig. 2(a). **c**, FC $B/H$ maps under the same stress conditions demonstrate a far weaker response, as expected in field cooling conditions.

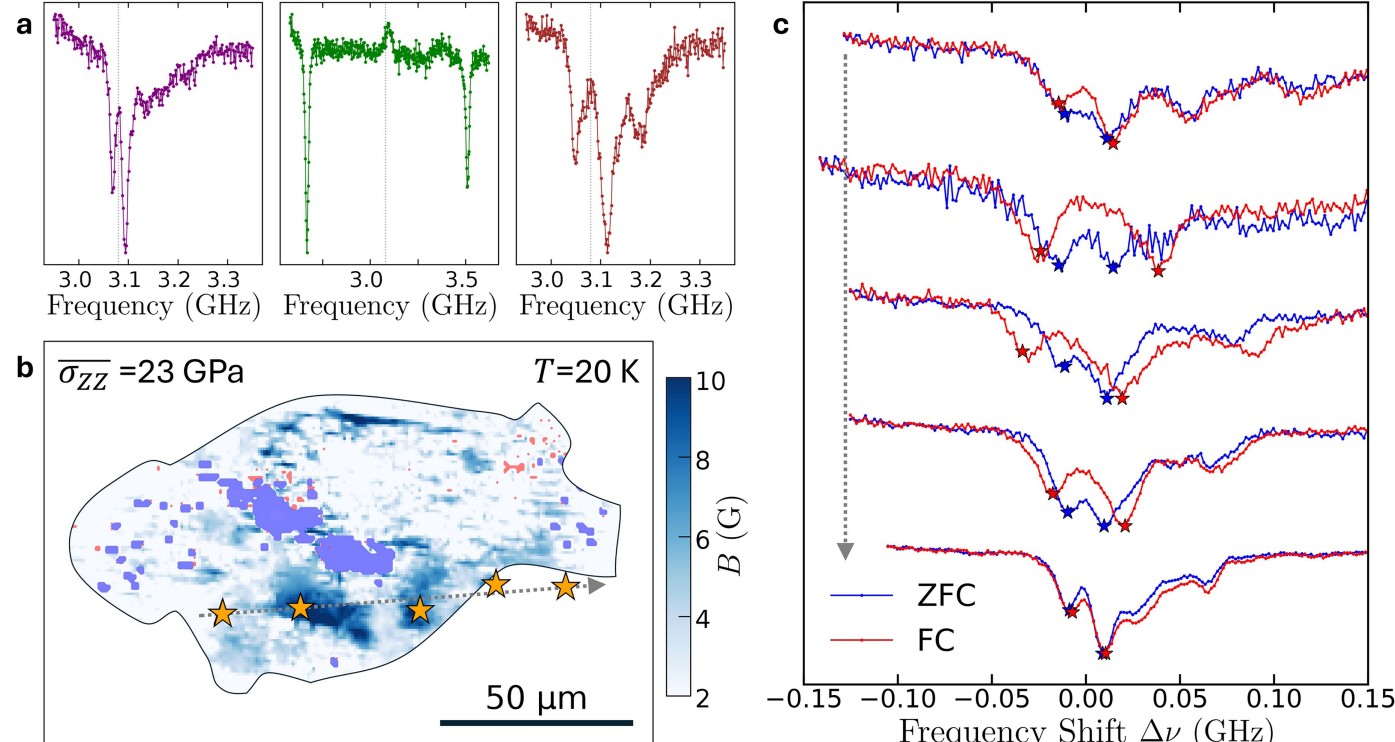

**Extended Data Fig. 7 | Flux trapping analysis in S1. a**, NV ODMR spectra of a chosen pixel after ZFC to 20 K (left), field ramping up to 150 G (middle), then FW up to 150 K and FC to 20 K and finally a field quench to 0 G (right). The dashed line indicates the shared $D_0 \approx 3.08$ GHz of the $NV^{(0)}$ group obtained from the middle spectrum. **b**, The map of trapped magnetic flux at 20 K and 23 GPa ($\overline{\sigma_{ZZ}}$).

Purple indicates pixels where low field ODMR spectra did not have sufficiently identifiable peaks. Red indicates pixels where the apparent $E_0$ is *smaller* after the flux trapping protocol and the $E_0$ cannot be interpreted as trapped flux. **c**, ODMR spectra of starred pixels along a line-cut in (**b**).

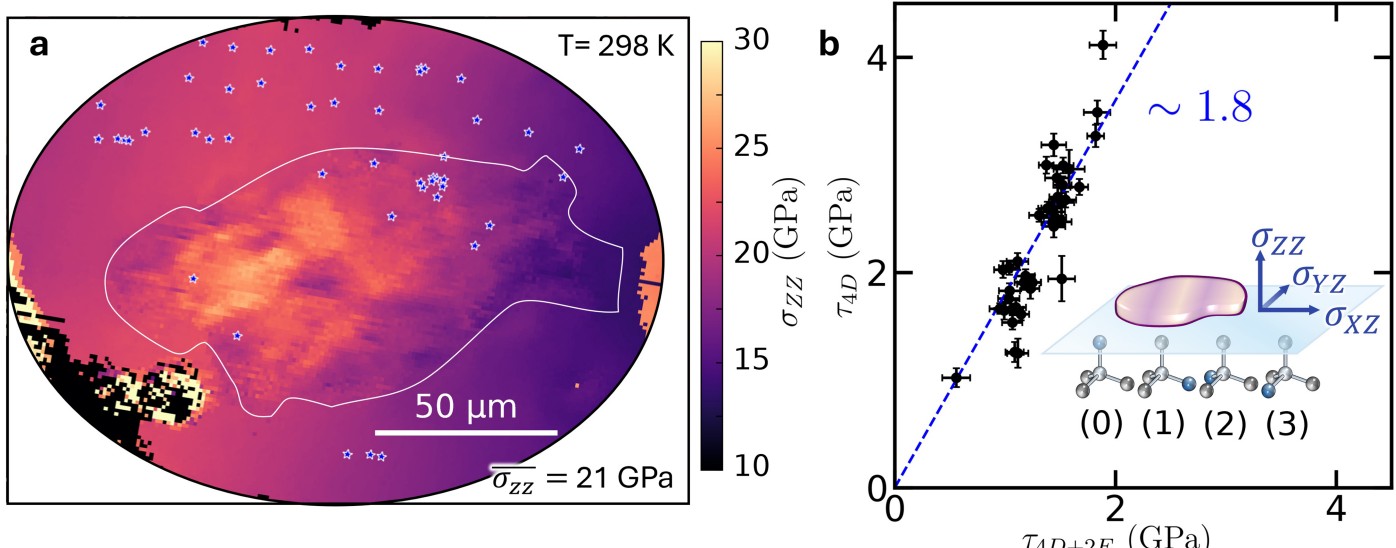

**Extended Data Fig. 8 | Correlation between two different shear reconstruction methods. a**, The $\sigma_{ZZ}$ map measured at $T = 298$ K and $\overline{\sigma_{ZZ}} = 21$ GPa which is identical between the two shear stress measurements (4D method and 4D+2E method). **b**, Scatter plot of shear stress calculated from the 4D method and shear stress calculated from the 4D+2E method with a fitted line $\tau_{4D} = (1.8 \pm 0.2)\tau_{4D+2E}$. The black dots correspond to randomly chosen pixels as blue stars in (**a**). The x (y) error corresponds to fitting error originated from peak splitting (peak width) uncertainties. The inset is a schematic of the three elements of the stress tensor $\sigma_{*z}$ continuous across the diamond plane, and the four labeled NV orientation subgroups.

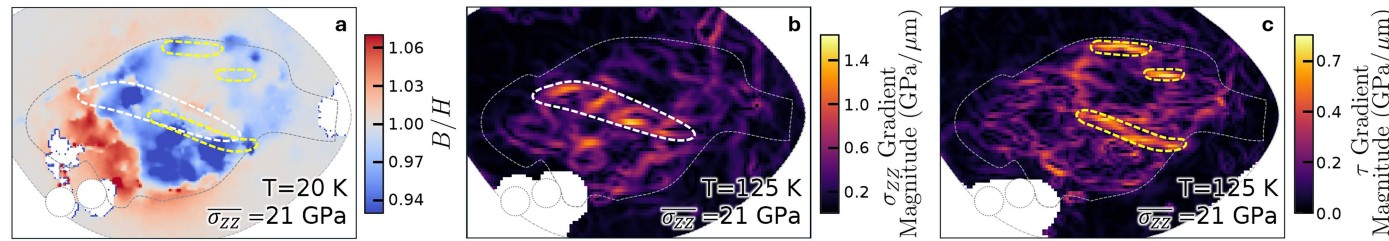

**Extended Data Fig. 9 | Comparison of the map of diamagnetism with the maps of the normal stress and shear stress gradients. a**, Map of diamagnetism in S1 at $\overline{\sigma_{ZZ}}$ = 21 GPa, with the highest regions of normal stress gradients and shear stress gradients marked in white and yellow, respectively. **b**, Map of normal stress ($\sigma_{ZZ}$) gradients. The region with the greatest gradient magnitude is encircled with a white dashed line. **c**, Map of shear stress ($\tau$) gradients. The region with the greatest gradient magnitude is encircled with a yellow dashed line.

**Extended Data Table 1 | Average *B/H* in S1, S2, and S3 across the sample chamber and superconducting region for different pressures and temperatures**

| | S1 | | | | | | | |
|---|---|---|---|---|---|---|---|---|
| | Sample Chamber | | | | Superconducting Region | | | |
| $\Sigma_i \frac{B_i}{NH}$ | 19 | 21 | 23 | 25 | 19 | 21 | 23 | 25 |
| 101 K | 1.003 | 0.998 | 1.000 | 1.000 | / | / | / | / |
| 77 K | 1.002 | 1.000 | 1.000 | 0.999 | / | / | / | / |
| 58 K | 1.002 | 0.999 | 1.000 | 1.001 | / | 0.972 | 0.991 | 0.985 |
| 35 K | 0.999 | 0.993 | 1.000 | 1.002 | 0.977 | 0.964 | 0.973 | 0.979 |
| 20 K | 1.002 | 1.000 | 1.000 | 1.002 | 0.983 | 0.964 | 0.972 | 0.978 |

| | S2 | | | | | |
|---|---|---|---|---|---|---|
| | Sample Chamber | | | Superconducting Region | | |
| $\Sigma_i \frac{B_i}{NH}$ | 18 | 23 | 27 | 18 | 23 | 27 |
| 51 K | 1.000 | 1.000 | 1.001 | / | / | / |
| 32 K | 0.999 | 1.001 | 1.001 | 0.984 | 0.989 | 0.986 |
| 20 K | 1.000 | 1.002 | 1.002 | 0.970 | 0.981 | 0.982 |

| | S3 | | | | | | | |
|---|---|---|---|---|---|---|---|---|
| | Sample Chamber | | | | Superconducting Region | | | |
| $\Sigma_i \frac{B_i}{NH}$ | 7.5 | 10 | 16 | 20 | 7.5 | 10 | 16 | 20 |
| 100 K | 0.999 | 1.000 | 1.000 | 1.001 | / | / | / | / |
| 10 K | 1.000 | 1.001 | 1.000 | 1.000 | / | / | 0.992 | 0.987 |

The slash (/) indicates points below the critical pressure or above the superconducting transition temperature, where the sample has not yet transitioned into the superconducting state. The same dome which is seen in Main Text Fig. 2(a) is again visible here in the table for S1 in the 'Superconducting Region' cells, now wrapped up as a single quantity, $\Sigma_i \frac{B_i}{NH}$. This quantity should be approximatley equal to 1, as indeed we see, when taken over the sample chamber.