## [Peer Review file · Nature]

Uncovering origins of heterogeneous superconductivity in La₃Ni₂O₇

Corresponding Author: Professor Norman Yao

Version 0:

Reviewer comments:

Referee #1

(Remarks to the Author)

Nickelates have emerged as a promising platform for investigating unconventional superconductivity, with the existence of bulk superconductivity in materials such as La₃Ni₂O₇ remaining one of the field's most significant unresolved questions. This work presents a groundbreaking investigation that combines quantum sensing techniques for magnetic field and stress measurements with detailed material composition analysis through EDX to address this fundamental puzzle. One of the study's most notable achievement is the observation of flux trapping—a crucial piece of evidence that has been absent from all previous investigations. Through systematic correlation of pressure/stress distribution and chemical composition with superconducting properties, the authors provide answers to questions troubled people in this field for so long. This work stands out as the most elegant and comprehensive quantum sensing study of nickelate superconductivity to date. Its systematic approach resolves numerous longstanding puzzles and uncertainties in the field. The thorough methodology and groundbreaking findings make this research of significant interest to the broader condensed matter physics community. I recommend its publication in Nature after the authors address the following questions.

1. In Fig.2e, it seems to me the largest flux trapping areas have the largest diamagnetic response in the ZFC-FW measurements. This is not intuitive to me because I imagine that those defect sites should trap the most flux, thus lead to the smallest diamagnetic response.
2. In Fig.2a, the spatial distribution of paramagnetism and diamagnetism are highly nonsymmetric. For example, at 20k, 21GPa, one side is paramagnetic, the other size is diamagnetic. Could the authors comment? If one sums up total flux, some plots does not seem to be right. One can even see in some plots, there are only paramagnetic areas in lower temperature cases. This does not seem to be right in physics.
3. The authors observe non-zero resistance and very small diamagnetism. Could the authors comment on the volume ratio of superconductor in the sample?
4. There are many mistakes in the reference list with some duplicated citations.

Referee #2

(Remarks to the Author)

In this work, the authors claim the three-dimensional 3D phase diagram of single-crystal La₃Ni₂O₇ as a function of temperature, normal stress/physical pressure, and shear stress by measuring diamagnetic signal using the NV quantum sensors. Their experimental results more clearly and directly reflect the superconducting state characteristics of single-crystal La₃Ni₂O₇, which can be an important step forward on the basis of all the present reported investigations. Their experimental results basically support that the superconductivity of the recently hotly studied nickle-based 327 compounds comes from the La₃Ni₂O₇ phase, which has certain research significance for future in-deep studies. The findings of this work are consistent conclusions have been made based on the study on the La₂PrNi₂O₇ thin films and poly-crystalline.

But there are some issues with the shear stress technique and the samples themselves, which may be the current conclusion that appears reliable but actually has significant flaws. The first one is that the La₃Ni₂O₇ sample itself has non-uniformity, with a symbiotic phase of 214 and 4310 in a ratio of approximately 1:1, which may weaken the reliability of the present high pressure experiment itself. Moreover, the superconductivity of La₃Ni₂O₇ under pressure is an intrinsic non-uniformity, and the superconducting volume is relatively small. Therefore, it is difficult to determine whether shear stress is the cause of non-bulk superconductors, but rather due to some chemical phase separation and O content that prevents superconductivity under high pressures. The second one is the evaluation of sheared samples. The average uni-axial stress of the entire sample is crucial for obtaining stress distributions across the sample, but equally important are the maximum and minimum shear stresses under high pressure. Especially for the uniaxial pressure of diamond pressure cell, the pressure gradient can be very large, so the shear stresses may vary greatly with different positions of the sample and diamond, which may lead to some deviations in the study of shear stresses.

This work may provide some potentially useful information and reports, may have considerable research significance, I thus agree that their results should be published in other journals as soon as possible, but their importance of the present article not truly meet the natural standards of nature although. It seems difficult to stimulate more relevant experiments and theoretical experiments, although the images of this work may be more clear than all previous results, and other more specialized journals may be more appropriate, including the nature communication and other. Considering this, my comments are as follows, which may be helpful to further improve the quality of the article.

1. The present results are relatively clear than all the previous investigations and the obtained results are positive and meaningful. But there are shortcomings, such as the fact that the non bulk superconducting state under high pressure, is not caused by pressure gradient or shear stress, and is closely related to specific sample quality and O content; Suggest the author to choose a relatively uniform system, such as the La₃PrNi₂O₇ single crystal and polycrystalline. This material doesn't exhibit bulk superconductivity under uniaxial pressure (DAC), and no superconducting transition was observed in magnetic susceptibility and electrical transport. However, perfect diamagnetism and superconductivity can be observed under hydrostatic pressure; Currently, it is widely believed that the difference in pressure gradient is a key factor. On this basis, studying the changes in superconducting states detected by NV quantum sensors may provide more reliable evidence.

2. Regarding the estimation of shear stress, the maximum, minimum, and average values are equally important. It is recommended that the author consider it, although the final results of the influence of shear stress on superconductivity are basically consistent with this article.

3. Considering the limitations of NV color centers itself and the shear stress, the excessive prospects in the article seem somewhat vague, for example, "understanding the microscopic mechanism by which shear stress destroys superconductivity in La₃Ni₂O₇ is a salient open question that may also shed light on recent studies of strained, thin-film-nickelates..." and "plastic deformations which generate defects, such as dislocations, that strongly affect the superconducting transition....", etc

Referee #3

(Remarks to the Author)

In this manuscript, the authors utilize arrays of NV centers embedded in diamond anvil cells to image the stress components and diamagnetic responses of bulk bilayer nickelate superconductors and correlate the results with transport and stoichiometry characterizations. They construct a compression, shear stress, and temperature phase diagram and highlight the necessity of correct La:Ni ratio for superconductivity. This experiment is done carefully and presented clearly. The results help provide a detailed view of the inhomogeneity in superconducting bulk samples and carry implications on how to further improve the superconductivity in bilayer nickelates. In this sense, this manuscript is of immediate interest to the nickelate superconductivity community. In addition, the demonstration of correlating transport, magnetic, stress, and stoichiometry measurements under high pressure is impressive and would appeal to the broad high-pressure research community.

On the other hand, there has been another similar work posted since last year [Preprint at arxiv:2410.10275], which raises the bar for originality. Both works report the diamagnetic response from the superconductivity and constructed a stress-temperature phase diagram outlining the superconducting "dome". In comparison, the current manuscript provides a more rigorous characterization of the stress condition with considerably higher statistics as well as an additional stoichiometry measurement. Specifically, the pixel registration reveals the detrimental effect of shear stress and off-stoichiometry to the superconductivity. Therefore, I in principle support the publication of this manuscript if the authors could address the following concerns:

1) Could the authors comment on the size of the diamagnetic response observed? The diamagnetic response in this report are all on the order of ~5% or lower despite the local nature of the NV center measurement as well as a considerably larger resistance drop seen in Extended Data Fig. 3. This is in clear contrast to the BSCCO benchmark shown in Extended Data Fig. 2 and different from more recent reports using samples also from M. Wang's group where superconducting volume above 40% is claimed (e.g. "Identification of Superconductivity in Bilayer Nickelate La₃Ni₂O₇ under High Pressure up to 100 GPa" <https://academic.oup.com/nsr/advance-article/doi/10.1093/nsr/nwaf220/8152906>).

Does this suggest that the superconductivity is related to the minority 1-3-1-3 phase of the La₃Ni₂O₇? Does this indicate low sample quality with, for example, significant amount of layer stacking faults which would not show up as off-stoichiometry? In

such a case, how robust are the authors' conclusions?

2) Could the authors make any statement on the sample alignment relative to the compression axis? Perhaps the authors could determine the crystal orientation after recovery from the high pressure cell, for example, of the sample shown in Extended Data fig. 4d? If the σ_{zz} and τ can be presented relative to the crystal axes, the phase diagram would be even more informative.

3) The ability of resolving the shear components from the total stress tensor is one of the highlights of this paper. However, the correlation between superconductivity and shear stress is not so clear. Why, in Fig. 4h, does the optimal superconductivity appear at non-zero shear stress? The left axis of the panel appears cut off at around 0.4 GPa too. Are there no data points near 0 shear stress? How is the panel f constructed then?

If one examines the correlation between Fig. 4d and e, there are spots colored in bright yellow in panel e that appears superconducting too. This also leads to the question of why the color scales for panel g and h are not set at max of 1.

4) Since the stoichiometry map is also 2D, could the authors construct a similar stoichiometry temperature phase diagram? It is rather strange how Fig. 5c only contains 8 data points with zero stoichiometry error bar. It would appear that a high-statistics statement on stoichiometry is possible here as well.

Some minor questions:

1) In Fig.4d, at some locations of sample S1, a maximum of >6% diamagnetic signal is observed. However, in Fig.4g and h, the largest suppression is ~3.5%. Why is there a discrepancy?

2) Regarding statistics and error bars, Fig. 5c suggests that there are appreciable errors in the B/H value which is not reflected in other figures, such as Fig. 5b, 3c, 2d.

3) This previous high pressure NV center work should be recognized and briefly discussed: arxiv:2410.10275.

Version 1:

Reviewer comments:

Referee #1

(Remarks to the Author)

The authors answered all questions satisfactorily. They added several new contents and boosted the quality of the manuscript further up. Despite one more paper on this topic popped out during the revision (PRL 135, 096001 (2025)), the importance of this work is not reduced, as it contains key physics missing in the other works, such as the origin of the inhomogeneity, the role of strain/stress/chemical composition, the proof of flux trapping and the estimation of superconducting ratio in the sample. Without these information, the field is still in the dark. Therefore, I recommend its publication in Nature.

Referee #2

(Remarks to the Author)

In the revised article, the authors have provided corresponding responses to all the comments raised; I also checked the evaluations or comments on the article from other reviewers, and they seem to suggest publishing this work in the nature journal. The experimental results clearly reflect superconducting state of single-crystal La₃Ni₂O₇ as well as the possible mechanism of filamentous superconductivity, which can be very important to the development of nickel-based superconductors. Nevertheless, I still believe that the present experimental results are not innovative enough, although the authors emphasized the importance and cutting-edge nature of NV quantum sensor technology. From the scientific perspective, I totally agree that this article is worth publishing as soon as possible since there have been some recent experimental reports measuring diamagnetic signal of La₃Ni₂O₇ by using the NV quantum sensors (eg., Phys. Rev. Lett. 135, 096001(2025); Natl Sci Rev, 12, nwaf 268 (2025), etc). Finally, I believe that the present manuscript may be on the brink of acceptance and requires the editor in the nature journal to comprehensively consider various factors to make the final conclusion.

Referee #3

(Remarks to the Author)

The authors have sufficiently addressed my questions. Although I now support its publication, I do note that the recent thin film development has marginalized the significance of this manuscript's results.

We thank all three referees for their extremely careful reading of our manuscript, and for the tremendously helpful suggestions that they have offered. We are thrilled by the overall positive responses from all three referees, with Referee 1 stating: “This work stands out as the most elegant and comprehensive quantum sensing study of nickelate superconductivity to date. Its systematic approach resolves numerous longstanding puzzles and uncertainties in the field. The thorough methodology and groundbreaking findings make this research of significant interest to the broader condensed matter physics community.” Referee 2 offers a similar sentiment regarding the potential impact on the broader community, explaining that: “Their experimental results more clearly and directly reflect the superconducting state characteristics of single-crystal $\text{La}_3\text{Ni}_2\text{O}_7$... and support that the superconductivity of the recently hotly studied nickel-based 327 compounds comes from the $\text{La}_3\text{Ni}_2\text{O}_7$ phase, which has certain research significance for future in-deep studies.” Finally, Referee 3 kindly highlights the breadth and complementarity of our manuscript: “This experiment is done carefully and presented clearly. The results help provide a detailed view of the inhomogeneity in superconducting bulk samples and carry implications on how to further improve the superconductivity in bilayer nickelates. In this sense, this manuscript is of immediate interest to the nickelate superconductivity community. In addition, the demonstration of correlating transport, magnetic, stress, and stoichiometry measurements under high pressure is impressive and would appeal to the broad high-pressure research community.”

In addition to these positive assessments, the referees raise a number of important questions and very helpful suggestions, which have greatly aided us in clarifying and sharpening the physical claims of our manuscript. We have carefully addressed all of their points in detail, and have revised our manuscript accordingly. We hope that these revisions as well as our responses below will allow all three referees to recommend publication of our manuscript in Nature.

I. REFEREE #1

Nickelates have emerged as a promising platform for investigating unconventional superconductivity, with the existence of bulk superconductivity in materials such as $\text{La}_3\text{Ni}_2\text{O}_7$ remaining one of the field's most significant unresolved questions. This work presents a groundbreaking investigation that combines quantum sensing techniques for magnetic field and stress measurements with detailed material composition analysis through EDX to address this fundamental puzzle. One of the study's most notable achievements is the observation of flux trapping—a crucial piece of evidence that has been absent from all previous investigations. Through systematic correlation of pressure/stress distribution and chemical composition with superconducting properties, the authors provide answers to questions troubled people in this field for so long. This work stands out as the most elegant and comprehensive quantum sensing study of nickelate superconductivity to date. Its systematic approach resolves numerous longstanding puzzles and uncertainties in the field. The thorough methodology and groundbreaking findings make this research of significant interest to the broader condensed matter physics community. I recommend its publication in Nature after the authors address the following questions.

We thank referee 1 for their careful reading of our manuscript and for their excellent summary of our key results. We very much appreciate their kind and enthusiastic assessment, in addition to their insightful comments and questions below. These suggestions have been very helpful for clarifying our own understanding of the data, and we have endeavored to respond to them thoroughly below.

1. In Fig.2e, it seems to me the largest flux trapping areas have the largest diamagnetic response in the ZFC-FW measurements. This is not intuitive to me because I imagine that those defect sites should trap the most flux, thus lead to the smallest diamagnetic response.

We thank the referee for bringing up this excellent observation. We agree that this point is rather counterintuitive and will try to explain our understanding below. Let us begin by considering a few limits regarding the behavior of superconductors as one increases the density of defects. As the referee suggests, a pristine type-II superconductor will develop a uniform vortex lattice for magnetic fields above H_{c1} . The introduction of defects creates pinning centers, and as the magnetic field decreases to zero, part of the magnetic flux remains trapped in the sample because the pinning energy of the defects exceeds the available thermal excitation energy. In this regime, the referee is absolutely correct that one expects the amount of trapped flux to anti-correlate with material

quality.

However, at a high enough defect concentration, the superconducting state will be destroyed and flux trapping is no longer possible at all. In this limit, the density of defects is no longer the limiting factor for flux trapping; rather, it is the presence of local regions of superconductivity in the first place which sets the ability to trap flux. As explored and characterized by many prior studies (as well as our own), the superconductivity seen in $\text{La}_3\text{Ni}_2\text{O}_7$ exhibits a relatively low volume fraction on the order of a few percent. Thus, we generally expect all superconducting regions of the sample to be defect-saturated enough to pin flux, and the amount of observed flux trapping is simply a proxy for the amount of superconductor present in that region. This heterogeneous limit is consistent with our data: for example, in sample S1, we observe a “donut” pattern in the flux trapping that mirrors the superconducting regions observed in ZFC-FW (zero-field cooling and then field warming). This reflects our understanding that if a region of the sample simply does not exhibit superconductivity, then it will also not exhibit flux trapping. Then, as the referee highlights, we also observe more flux trapped in regions with the largest diamagnetic response. This is again consistent with our understanding that the sample is extremely heterogeneous: these large-diamagnetic-response regions simply contain more superconductor, which enables them to trap more flux per unit volume. We have added this important discussion to our manuscript.

2. In Fig.2a, the spatial distribution of paramagnetism and diamagnetism are highly nonsymmetric. For example, at 20 K, 21GPa, one side is paramagnetic, the other side is diamagnetic. Could the authors comment? If one sums up total flux, some plots does not seem to be right. One can even see in some plots, there are only paramagnetic areas in lower temperature cases. This does not seem to be right in physics.

We thank the referee for this extremely insightful question and for observing a very salient and striking feature of our diamagnetism maps. To be maximally honest, the authors spent a lot(!) of time trying to understand and model our observed spatial distribution of paramagnetic and diamagnetic regions. We are still not sure we totally understand the underlying microscopic origins, but will try our best to explain to the referee our current hypothesis. On the other hand, perhaps even more importantly, motivated by the referee’s question, we have gone back and carefully checked that the sum of the total magnetic flux across the sample chamber does *indeed* add up to the correct external applied field.

S1								
	Sample Chamber				Superconducting Region			
$\Sigma_i \frac{B_i}{NH}$	19 GPa	21 GPa	23 GPa	25 GPa	19 GPa	21 GPa	23 GPa	25 GPa
101 K	1.003	0.998	1.000	1.000	/	/	/	/
77 K	1.002	1.000	1.000	0.999	/	/	/	/
58 K	1.002	0.999	1.000	1.001	/	0.972	0.991	0.985
35 K	0.999	0.993	1.000	1.002	0.977	0.964	0.973	0.979
20 K	1.002	1.000	1.000	1.002	0.983	0.964	0.972	0.978

S2						
	Sample Chamber			Superconducting Region		
$\Sigma_i \frac{B_i}{NH}$	18 GPa	23 GPa	27 GPa	18 GPa	23 GPa	27 GPa
51 K	1.000	1.000	1.001	/	/	/
32 K	0.999	1.001	1.001	0.984	0.989	0.986
20 K	1.000	1.002	1.002	0.970	0.981	0.982

S3								
	Sample Chamber				Superconducting Region			
$\Sigma_i \frac{B_i}{NH}$	7.5 GPa	10 GPa	16 GPa	20 GPa	7.5 GPa	10 GPa	16 GPa	20 GPa
100 K	0.999	1.000	1.000	1.001	/	/	/	/
10 K	1.000	1.001	1.000	1.000	/	/	0.992	0.987

TABLE I: Average B/H in S1, S2, and S3 across the sample chamber and superconducting region for different pressures and temperatures. The slash (/) indicates points below the critical pressure or above the superconducting transition temperature, where the sample has not yet transitioned into the superconducting state. The same dome which is seen in the main Fig. 2(a) is again visible here in the table for S1 in the "Superconducting Region" cells, now wrapped up as a single quantity, $\Sigma_i \frac{B_i}{NH}$.

Sum of total flux consistency check: As the referee mentions, magnetostatics dictates that the total magnetic flux through the entire sample chamber should remain the same, regardless of whether a sample is present or not. That is to say, $\Sigma_i B_i \approx NH$, where the index i runs over all N pixels in a spatial map, B_i is the field measured at a given pixel, and H is the applied field. Equivalently, $\Sigma_i \frac{B_i}{NH} \approx 1$. To this end, we explicitly calculate the value of $\Sigma_i \frac{B_i}{NH}$ (across the entire sample chamber) for all three samples and depict the values for Sample S1 in Table I above (we note that this "flux check" table for all three samples is now shown in the Methods). As the referee will see, for all temperatures and pressures, the value of $\Sigma_i \frac{B_i}{NH}$ is extremely close to unity. To ensure that this is not somehow just a quantitative artifact, we also compute $\Sigma_i \frac{B_i}{NH}$ over just the superconducting regions and observe values consistent with a $\sim 2-4\%$ diamagnetic suppression in the superconducting region. We have added the relevant tables and the above discussion for all samples to the Methods.

Spatial distribution of paramagnetic and diamagnetic regions: As we have alluded to above, we do not believe that we have a full microscopic understanding of the origin of the observed spatial distribution of paramagnetic and diamagnetic regions in our samples. In fact, we think that understanding this in detail represents one of the most intriguing future directions to undertake. To

this end, we apologize for the length and detail of our response below, but we will try our best to explain to the referee our current “world view.”

To begin, let us remind ourselves what a perhaps “normal” diamagnetic map for a clean superconductor should look like. As depicted in Fig. 2 of the Extended Data, when one looks at the magnetic field maps of BSCCO, one finds that $B/H < 1$ occurs directly below the sample, while $B/H > 1$ occurs at the rim of the sample. This aligns with what we expect: the Meissner effect displaces fields from the sample’s bulk and the regions of “enhancement” ($B/H > 1$) are reasonably symmetric around the BSCCO sample. As the referee points out, this is not at all true for our $\text{La}_3\text{Ni}_2\text{O}_7$ samples; for example, for S1, enhancement tends to appear only on one side of the sample.

In the main text, we very briefly commented on this phenomenon as possibly being related to the so-called Wohleben effect (aka the Paramagnetic Meissner Effect). We will now explain this understanding in quite a bit more detail and hope that the referee appreciates this additional information. As the referee notes, there seems to be two observations that are in a bit of tension: first, the paramagnetic enhancement seems to be somewhat disconnected from the superconducting regions that exhibit Meissner suppression. On the other hand, we observe that the regions of enhancement share a common T_c with the regions of Meissner suppression, so this phenomena is likely to still be linked to superconductivity. This type of behavior — i.e. a superconductor developing regions of paramagnetism below T_c — has been extensively observed and documented in *granular* superconductors, where it goes under the moniker of the Wohleben effect. For example, this effect has been observed in the nickelate superconductor $\text{Nd}_{0.8}\text{Sr}_{0.2}\text{NiO}_2$, in cuprates and other classes of unconventional superconductors [1–4], as well as in certain BCS superconductors such as Nb, Al, and Pb[5]. After decades of study, the underlying microscopic physics is still in debate, but a common theme seems to be that strong heterogeneity or granularity is a general prerequisite. For example, some models rely on Josephson coupling between discrete superconducting grains leading to spontaneous currents, which generate the paramagnetism. Other models also require d-wave superconductivity [6], leading to the Wohleben effect being proposed as a method to probe the symmetry of the superconducting order parameter. The fact that granularity seems to be crucial for observing this type of paramagnetic enhancement seems to point in the right direction for our $\text{La}_3\text{Ni}_2\text{O}_7$ samples, which exhibit strong heterogeneity at the micron scale.

Finally, although the Wohleben effect has most often been examined under field-cooled (FC) protocols in prior works, some studies report a small paramagnetic signal in zero-field-cooled (ZFC)

measurements a few kelvin below T_c [6]. In contrast, we observe a clear ZFC paramagnetic response across the entire sub- T_c range, with a magnitude that is approximately proportional to the diamagnetic signal. We are (and have been for a while) in active discussions with theorists to better understand this phenomenon. Zooming out a bit, at a high level, some version of the Wohleben effect seems like the most reasonable explanation for the data we see, given its robustness across pressure points and samples and its appearance in our “granular” $\text{La}_3\text{Ni}_2\text{O}_7$ system and not in other clean systems that we have studied such as BSCCO. We have added a more extensive discussion about the regions of paramagnetic enhancement to our manuscript.

3. The authors observe non-zero resistance and very small diamagnetism. Could the authors comment on the volume ratio of superconductor in the sample?

We thank the referee for this excellent suggestion! Indeed, as the referee highlights, a very important metric for the evaluation of nickelate superconductivity has been the volume fraction as deduced via magnetic susceptibility measurements. Crucially, our local measurements of B/H provide an alternate route to extract the superconducting volume fraction of the sample.

Our NV centers (embedded in the anvil culet) are ≈ 500 nm from the sample, which is significantly smaller than the sample dimensions ($\approx 10 - 100$ μm). Thus, we treat the NVs as measuring the stray field at the surface of the sample. In order to extract the superconducting volume fraction f from our experimentally measured average diamagnetic suppression, $\langle B/H \rangle$, we work with the following simple model. Assume that a fraction f of the sample is a perfect superconductor with $\chi_{SC} = -1$, and the remaining fraction of the material has a paramagnetic response i.e $\chi_{PM} \sim 0$. The effective magnetic susceptibility of the sample is then given by:

$$\chi_{eff} = f\chi_{SC} + (1 - f)\chi_{PM} \sim -f \quad (1)$$

Since our NVs effectively measure the stray field at the surface of the sample, we can approximate $\chi_{eff} \sim \langle B/H \rangle - 1$ and therefore $|\chi_{eff}| \sim 1 - \langle B/H \rangle$. Thus, for our three samples, we obtain a superconducting volume fraction of: 3%, 0.3%, 0.5% for S1, S2 and S3, respectively. We note that these values are relatively consistent with prior measurements from AC susceptibility using a solid/liquid pressure transmitting medium [7].

To validate our estimates above, which make use of the stray field at the surface of the sample, we numerically simulate a heterogeneous superconducting slab of dimensions $(L_x, L_y, L_z) =$

$(128\mu m, 128\mu m, 8\mu m)$ placed in an external magnetic field $\vec{H} = 100G \hat{z}$.

We assume that the slab is composed of $1\mu m$ -sized superconducting grains, which are randomly distributed at a fixed superconducting volume fraction. We model the superconducting grains as perfect dipoles whose response to an external magnetic field, H , is perfectly diamagnetic. From our simulations, we find that for low superconducting volume fractions, there is a direct linear relationship between $\langle B/H \rangle$ (measured from the stray field at the surface of the sample) and the superconducting volume fraction, consistent with our analytic assumptions above. We thank the referee again for this suggestion and have added a new section to the Methods, detailing the extraction of the superconducting volume fraction from measurements of our B/H data.

4. There are many mistakes in the reference list with some duplicated citations.

We thank the referee for pointing this out and have fixed all of these issues.

FIG. R1: Simulation geometry for evaluating $\langle B/H \rangle$ at a given superconducting volume fraction. All dimensions, including sample size and NV depth, correspond to true experimental geometries.

II. REFEREE #2

In this work, the authors claim the three-dimensional 3D phase diagram of single-crystal $\text{La}_3\text{Ni}_2\text{O}_7$ as a function of temperature, normal stress/physical pressure, and shear stress by measuring diamagnetic signal using the NV quantum sensors. Their experimental results more clearly and directly reflect the superconducting state characteristics of single-crystal $\text{La}_3\text{Ni}_2\text{O}_7$, which can be an important step forward on the basis of all the present reported investigations. Their experimental results basically support that the superconductivity of the recently hotly studied nickel-based 327 compounds comes from the $\text{La}_3\text{Ni}_2\text{O}_7$ phase, which has certain research significance for future in-deep studies. The findings of this work are consistent conclusions have been made based on the study on the $\text{La}_2\text{PrNi}_2\text{O}_7$ thin films and poly-crystalline.

We thank the referee for their detailed reading of our manuscript, as well as their very insightful suggestions. Many of the referees questions, such as the role of different $\text{La}_3\text{Ni}_2\text{O}_7$ phases have been extremely helpful and have allowed us to more carefully explain the reasoning behind our conclusions. These revisions have also helped to improve the clarity of our manuscript.

But there are some issues with the shear stress technique and the samples themselves, which may be the current conclusion that appears reliable but actually has significant flaws. The first one is that the $\text{La}_3\text{Ni}_2\text{O}_7$ sample itself has non-uniformity, with a symbiotic phase of 214 and 4310 in a ratio of approximately 1:1, which may weakens the reliability of the present high pressure experiment itself.

Moreover, the superconductivity of $\text{La}_3\text{Ni}_2\text{O}_7$ under pressure is an intrinsic non-uniformity, and the superconducting volume is relatively small. Therefore, it is difficult to determine whether shear stress is the cause of non-bulk superconductors, but rather due to some chemical phase separation and O content that prevents superconductivity under high pressures.

We completely agree with the referee that high-pressure nickelate superconductivity is sensitive to a multitude of different types of inhomogeneities, including the presence of 214/4310 phases, as well as variations in oxygen content. We also agree that it is absolutely crucial to rule out these other *intrinsic* sources of inhomogeneity at the material level, in terms of explaining our observations. To this end, motivated by the referee's question, we have gone back and more carefully examined our spatial diamagnetic maps as a function of pressure, with the goal of understanding whether an intrinsic chemical or structural inhomogeneity could in fact be the origin of our observations.

In particular, we focused on pressures near the optimal pressure and attempted to systematically

FIG. R2: **a** Map of diamagnetism reproduced from Main Text Fig. 2a at $\overline{\sigma_{zz}} = 19$ GPa. **b** Map of diamagnetism reproduced from Main Text Fig. 2a at $\overline{\sigma_{zz}} = 21$ GPa. Together, these maps highlight how diamagnetism in the sample's center is quenched upon increasing pressure. The central region encircled by a grey dashed line in each subfigure indicates the region for which this behavior is most prominent. Just as in the main text, the images have been aligned via pixel-registration.

track changes in the regions of the sample which exhibit a Meissner effect. As depicted in Fig. R2(a), at $\overline{\sigma_{zz}} = 19$ GPa, we find that the central region of the sample (gray dashed line) exhibits fairly consistent diamagnetism with $B/H < 1$. However, at the next pressure point, $\overline{\sigma_{zz}} = 21$ GPa, there is no longer a measurable diamagnetic signal in that region of the sample. The fact that there are regions of superconductivity which appear at lower pressures, but which *disappear* as one increases the pressure cannot be easily explained by intrinsic heterogeneous variations of the material. Indeed, neither oxygen stoichiometry variations nor the presence of alternative chemical phases can explain this behavior. Moreover, at $\overline{\sigma_{zz}} = 19$ GPa, the fact that the center of the sample exhibits a Meissner response and thus a superconducting volume fraction similar to that of the edge of the sample, also suggests that intrinsic material differences are likely not the origin of the subsequently observed spatial variations in B/H at higher pressures.

By contrast, we respectfully emphasize that there is nearly a perfect *real space* overlap between regions of the shear stress map above a critical shear and regions of the diamagnetism map which do not exhibit superconductivity. This direct real space correlation (Fig. 4d,e in the maintext) is perhaps the strongest evidence for the fact that extrinsic shear is an important factor in controlling the regions of superconductivity observed in our samples. Of course, as explored in sample S3, we are certainly not claiming that shear is the *only* type of inhomogeneity that can matter and chemical inhomogeneities, when they exist, are certainly expected to impact the sample's local superconducting response.

The second one is the evaluation of sheared samples. The average uni-axial stress of the entire

FIG. R3: Histograms of the shear stress across the sample over the stress points where the full stress tensor was collected. The inset shows the sample region at each pressure, over which the histogram was taken, with a color indicating the shear stress magnitude, τ , in gigapascals (GPa). The minimum and maximum shear stresses at each stress point is marked on the inset with a blue and red X, respectively. This information is further displayed in Table II, which give single values for the minimum, maximum, and average shear stress magnitudes in gigapascals.

sample is crucial for obtaining stress distributions across the sample, but equally important are the maximum and minimum shear stresses under high pressure. Especially for the uniaxial pressure of diamond pressure cell, the pressure gradient can be very large, so the shear stresses may vary greatly with different positions of the sample and diamond, which may lead to some deviations in the study of shear stresses.

We thank the referee for bringing up this insightful suggestion. The referee is absolutely correct that there are pressure and shear gradients across the sample, and that it is important to characterize the minimum and maximum shear stresses. To this end, as shown in Fig. R3, we present a histogram of the shear stresses for sample S1 at three different pressures. This allows the referee to visualize both the minimum and maximum shears as well as the distribution of shears across the sample (also shown via spatial color maps in the inset of each histogram). As the referee suggests, in each case there is a broad distribution of shears across the sample.

Next, the referee brings up an extremely important and valid question, which is whether the presence of large *pressure gradients* might lead to shear variations at different positions that make it challenging to accurately extract the local shear stress. Let us address the excellent question in detail. Perhaps most crucially, we note that τ (shear stress) and σ_{zz} (uniaxial stress) are *independently* resolved by the NV center. In particular, because there are four crystallographic NV orientations, we are able to locally calculate σ_{zz} by comparing the resonances of the perpendicular-to-culet NV subgroup (D_0) with those of the three other NV subgroups (D_1, D_2, D_3), which are predominantly oriented in-plane. In contrast, a shear stress, τ , breaks the in-plane symmetry at a given point,

FIG. R4: Comparison of the map of diamagnetism with the maps of the normal stress and shear stress gradients. **a** Map of diamagnetism in S1 at $\overline{\sigma_{zz}} = 21$ GPa, with the highest regions of normal stress gradients and shear stress gradients marked in white and yellow, respectively. **b** Map of normal stress (σ_{zz}) gradients. The region with the greatest gradient magnitude is encircled with a white dashed line. **c** Map of shear stress (τ) gradients. The region with the greatest gradient magnitude is encircled with a yellow dashed line.

and this manifests as a completely different NV ODMR response. In particular, this appears as a *difference* in the values of (D_1, D_2, D_3) . Thus, much like prior studies which utilize the same methodology [8–10], we can be quite confident in our extraction of shear stresses *even when* there are concomitant pressure gradients.

Finally, given the broad distribution of both uniaxial and shear stresses, the referee alludes to an extremely interesting possibility: namely, is it possible that gradients of either the uniaxial or shear stress could in fact also correlate with the observed spatial inhomogeneity in the diamagnetic maps? To explore this possibility, as shown in Fig. R4, we plot the spatial gradients of both σ_{zz} and τ alongside the corresponding map of diamagnetism (for sample S1 at $\overline{\sigma_{zz}} = 21$ GPa). The largest uniaxial stress gradients are indicated by the dashed white line in Fig. R4b, while the largest shear gradients are indicated by the yellow dashed lines in Fig. R4c. Crucially, neither of these gradient maxima exhibit any meaningful correlated overlap with the non-diamagnetic region of the sample. This is, of course, in stark contrast to the direct spatial correlations of the non-Meissner region of the sample with large shear stresses above a critical value.

We thank the referee again for this very intriguing set of questions and have added this important exploration and analysis of shear and uniaxial stress gradients into our Methods.

This work may provide some potentially useful information and reports, may have considerable research significance, I thus agree that their results should be published in other journals as soon as possible, but their importance of the present article not truly meet the natural standards of nature although. It seems difficult to stimulate more relevant experiments and theoretical experiments, although the images of this work may be more clear than all previous results, and other more specialized journals may be more appropriate, including the nature communication and other.

We truly appreciate the referee's comments and are very grateful for their assessment that our results "may have considerable research significance". The referee's questions have led us to analyze multiple aspects of our data more carefully (e.g. role of shear gradients, spatial disappearance of diamagnetic signals upon increasing pressure, etc) and we believe that these additional analyses have strengthened our manuscript. To this end, we very much hope that this referee will agree with the other two referees that our manuscript may be published in Nature.

Considering this, my comments are as follows, which may be helpful to further improve the quality of the article.

1. The present results are relatively clear than all the previous investigations and the obtained results are positive and meaningful. But there are shortcomings, such as the fact that the non bulk superconducting state under high pressure, is not caused by pressure gradient or shear stress, and is closely related to specific sample quality and O content; Suggest the author to choose a relatively uniform system, such as the $\text{La}_3\text{PrNi}_2\text{O}_7$ single crystal and polycrystalline.

We thank the reviewer for this very interesting suggestion. We completely agree that a study of $\text{La}_2\text{PrNi}_2\text{O}_7$ would help to further augment our understanding of the family of Ruddelston-Popper nickelates. In fact, we have begun reaching out to our collaborators in an attempt to secure samples for such a study. However, to be maximally honest, the full explorations of a *single* $\text{La}_3\text{Ni}_2\text{O}_7$ sample from our current manuscript (i.e. any one of the samples S1, S2 and S3) takes approximately $\sim 4-6$ months to accumulate data for! Of course, this is partially owing to the tremendous amount of care we take in our measurements and analyzing our data, but this hopefully gives the referee a rough feeling for the time-scales associated with such careful studies. To this end, we very respectfully believe that a meaningful study of $\text{La}_2\text{PrNi}_2\text{O}_7$ is outside the scope of our current study.

This material doesn't exhibit bulk superconductivity under uniaxial pressure (DAC), and no superconducting transition was observed in magnetic susceptibility and electrical transport.

We are a bit confused by this point and must respectfully disagree with the referee. We are not precisely sure that we understand the referee's question so we will do our best to attempt to answer two possible interpretations of the referee's question.

- Interpretation 1: the referee is saying that under uniaxial stress, the material has not been observed to exhibit bulk superconductivity. We do not understand this question since no DAC experiment can achieve perfectly uniaxial, or even dominantly uniaxial pressures, up to tens of

GPa. Thus, there is no experiment that has explored or is even able to explore such a parameter regime.

- Interpretation 2: the referee is saying that by using a DAC, which has some excess uniaxial stress (but is dominantly hydrostatic), the material has not been observed to exhibit bulk superconductivity. Again, we believe that this is simply not correct. Numerous DAC studies have detected a superconducting transition in electrical transport and in magnetic susceptibility in $\text{La}_3\text{Ni}_2\text{O}_7$ [11–15]. Indeed, even the original report by Meng Wang observed signatures of superconductivity using a solid pressure transmitting media inside a DAC [11].

However, perfect diamagnetism and superconductivity can be observed under hydrostatic pressure; Currently, it is widely believed that the difference in pressure gradient is a key factor. On this basis, studying the changes in superconducting states detected by NV quantum sensors may provide more reliable evidence.

We thank the referee for this positive comment and certainly agree that our NV quantum sensors are able to provide more reliable and local evidence for superconducting behavior in $\text{La}_3\text{Ni}_2\text{O}_7$. At the same time, we would like to respectfully clarify that, to the best of our knowledge, there has been no clear microscopic explanation for why one should expect hydrostatic pressures to favor superconductivity in the nickelates. To this end, we believe that our study is quite unique in that it establishes, for the first time, shear as a key antagonistic factor for nickelate superconductivity; we hasten to emphasize that this microscopic understanding is only possible because of our novel metrological toolkit.

2.Regarding the estimation of shear stress, the maximum, minimum, and average values are equally important. It is recommended that the author consider it, although the final results of the influence of shear stress on superconductivity are basically consistent with this article.

We thank the referee for this suggestion, and have created a table including this information as shown below (Table II).

3.Considering the limitations of NV color centers itself and the shear stress, the excessive prospects in the article seem somewhat vague, for example, “understanding the microscopic mechanism by which shear stress destroys superconductivity in $\text{La}_3\text{Ni}_2\text{O}_7$ is a salient open question that may also shed light on recent studies of strained, thin-film-nickelates...” and “plastic deformations which

	$\overline{\sigma_{ZZ}} = 21$ GPa	$\overline{\sigma_{ZZ}} = 23$ GPa	$\overline{\sigma_{ZZ}} = 25$ GPa
Max Shear	3.49	3.65	4.22
Min Shear	0.05	0.08	0.11
Avg Shear	1.25	1.41	1.51

TABLE II: Shear stress statistics for each stress point. These numbers provide a summary of the more in-depth picture provided by the histograms displayed in Fig. R3. The locations at which the minimum and maximum shear values occur are marked on Fig. R3, while the average was taken across the full sample region.

generate defects, such as dislocations, that strongly affect the superconducting transition....”, etc

We very much appreciate the referee’s perspective here and agree that one must be very careful in not stretching claims of impact. To this end, we have carefully modified those sentences to more accurately and precisely express the scientific implications. Finally, we would like once again thank the referee for their helpful questions and suggestions. We hope that our new analyses and revisions to the manuscript, as well as our careful replies above, will enable this referee to agree with the other two referees that our manuscript may be published in Nature.

III. REFEREE #3

In this manuscript, the authors utilize arrays of NV centers embedded in diamond anvil cells to image the stress components and diamagnetic responses of bulk bilayer nickelate superconductors and correlate the results with transport and stoichiometry characterizations. They construct a compression, shear stress, and temperature phase diagram and highlight the necessity of correct La:Ni ratio for superconductivity. This experiment is done carefully and presented clearly. The results help provide a detailed view of the inhomogeneity in superconducting bulk samples and carry implications on how to further improve the superconductivity in bilayer nickelates. In this sense, this manuscript is of immediate interest to the nickelate superconductivity community. In addition, the demonstration of correlating transport, magnetic, stress, and stoichiometry measurements under high pressure is impressive and would appeal to the broad high-pressure research community.

On the other hand, there has been another similar work posted since last year [Preprint at arxiv:2410.10275], which raises the bar for originality. Both works report the diamagnetic response from the superconductivity and constructed a stress-temperature phase diagram outlining the superconducting “dome”. In comparison, the current manuscript provides a more rigorous characterization of the stress condition with considerably higher statistics as well as an additional stoichiometry measurement. Specifically, the pixel registration reveals the detrimental effect of shear stress and off-stoichiometry to the superconductivity. Therefore, I in principle support the publication of this manuscript if the authors could address the following concerns:

We thank the referee for their extremely careful reading of our manuscript and their excellent summary of our main results. We apologize for not providing a discussion of arXiv:2410.10275 and have now added this to the main text. The referee offers a number of very insightful questions and suggestions (e.g. a stoichiometry-temperature phase diagram), which have helped us to improve both the clarity and scientific impact of our manuscript.

1) Could the authors comment on the size of the diamagnetic response observed? The diamagnetic response in this report are all on the order of 5% or lower despite the local nature of the NV center measurement as well as a considerably larger resistance drop seen in Extended Data Fig. 3. This is in clear contrast to the BSCCO benchmark shown in Extended Data Fig.2 and different from more recent reports using samples also from M. Wang’s group where superconducting volume above 40% is claimed (e.g. “Identification of Superconductivity in Bilayer Nickelate $\text{La}_3\text{Ni}_2\text{O}_7$ under High Pressure up to 100 GPa” <https://academic.oup.com/nsr/advance-article/doi/10.1093/nsr/nwaf220/8152906>).

Does this suggest that the superconductivity is related to the minority 1-3-1-3 phase of the $\text{La}_3\text{Ni}_2\text{O}_7$? Does this indicate low sample quality with, for example, significant amount of layer stacking faults which would not show up as off-stoichiometry? In such a case, how robust are the authors' conclusions?

We thank the referee for this excellent question. In fact, in order to fully answer the referee's point, we had to do some more detailed investigations and analyses ourselves. We apologize for the length of the reply below, but we hope that the referee will appreciate our detailed thoughts.

As the referee mentions, the diamagnetic response that we observe in the various nickelate samples are on the order of $\sim 5\%$, while there have been recent reports of superconducting volume fractions of $\text{La}_3\text{Ni}_2\text{O}_7$ up to $\sim 40\%$. When we started trying to explore this question, we realized that directly relating the diamagnetic suppression measured by our NV centers with the superconducting volume fraction is actually a bit non-trivial. To this end, let us begin by describing how to quantitatively extract the sample's superconducting volume fraction f from our experimentally measured average diamagnetic suppression, $\langle B/H \rangle$.

From diamagnetic suppression to volume fractions: Our NV centers (embedded in the anvil culet) are ≈ 500 nm from the nickelate sample, which is significantly smaller than the sample dimensions ($\approx 10 - 100$ μm). To this end, our NVs are effectively measuring the stray field at the surface of the sample. In order to provide a quantitative mapping from the average diamagnetic suppression to the superconducting volume fraction, let us consider the following simple model. Assume that a fraction f of the sample is a perfect superconductor with $\chi_{SC} = -1$, and the remaining fraction of the material has a paramagnetic response i.e $\chi_{PM} \sim 0$. The effective magnetic susceptibility of the sample is then given by, $\chi_{\text{eff}} = f\chi_{SC} + (1 - f)\chi_{PM} \sim -f$. Since our NVs effectively measure the stray field at the surface of the sample, one would naturally expect that $\chi_{\text{eff}} \sim \langle B/H \rangle - 1$ and therefore $|\chi_{\text{eff}}| \sim 1 - \langle B/H \rangle$. To validate this expectation, which makes use of the stray field at the surface of the sample, we numerically simulate a heterogeneous superconducting slab of dimensions $(L_x, L_y, L_z) = (128\mu\text{m}, 128\mu\text{m}, 8\mu\text{m})$ placed in an external magnetic field $\vec{H} = 100\text{G} \hat{z}$. We assume that the slab is composed of $1\mu\text{m}$ -sized superconducting grains, which are randomly distributed at a fixed superconducting volume fraction, f . We model the superconducting grains as perfect dipoles whose response to an external magnetic field, H , is perfectly diamagnetic. From our simulations, we find that for low superconducting volume fractions, there is a direct linear relationship between $\langle B/H \rangle$ (measured from the stray field at the surface of the sample) and the

Volume fraction %			Pressure medium	Measurement method	year	ref
S1	S2	S3	Silicone oil	AC susceptibility, in situ coils	2025	[16]
0.68	0	0				
S1	S2	S3	Helium	DC susceptibility, in MPMS	2025	[12]
41	33	31				
S1	S2	S3	NaCl	NV stray field	2025	this work
3	0.3	0.5				

TABLE III: Summary of all manuscripts where quantitative superconducting volume fractions of $\text{La}_3\text{Ni}_2\text{O}_7$ are reported. Note that for Ref. [16], there are two additional samples (S4, S5) where a superconducting volume fraction of 0% is reported, but which we do not include in the table owing to space constraints. The type of pressure transmitting medium and measurement method are also indicated.

superconducting volume fraction, consistent with our analytic expectation above. Thus, as perhaps intuitively expected by this referee, it is indeed completely reasonable to compare the the average diamagnetic suppression measured from our samples with the superconducting volume fractions reported in other studies.

Origins of low superconducting volume fraction: Having established that the diamagnetic suppression provides a quantitative proxy for the superconducting volume, we now turn to the referee’s question to try to understand why our observed effective volume fractions are smaller than recent measurements from Meng’s group. We begin by just quickly reiterating what the referee has already mentioned, namely, that for our BSCCO benchmark experiments, we do in fact observe a near-total Meissner suppression. To be a bit more quantitative about our nickelate samples, we note that the largest diamagnetic signature is seen in S1, with $\langle B/H \rangle \approx 0.89$ suggesting a local superconducting volume fraction of approximately 11%. Moreover, within the superconducting regions of S1 at $\overline{\sigma_{zz}} = 21$ GPa and 20K, B/H varies roughly between 3 and 11 percent. Nevertheless, as the referee notes, these values are indeed significantly lower than those reported in Meng’s most recent manuscript. To be maximally frank, we wondered perhaps whether Meng’s synthesis methods had changed in recent years since our samples from his group were obtained as a part of their original growth batch. Meng’s response to our question was as follows:

“For the high-pressure floating zone technique, we did not change the growth process. We selected better samples based on high-pressure transport measurements and then measured

them with gas pressure-transmitting medium. We found that some samples can yield a SC volume above 40%.”

Following this response from Meng, we went back and carefully tabulated all the $\text{La}_3\text{Ni}_2\text{O}_7$ samples for which superconducting volume fractions have been reported. The results are shown in Table III above. As far as we are aware from Meng, all the $\text{La}_3\text{Ni}_2\text{O}_7$ samples used in the three studies indicated in the Table are grown using the same method and the measurements differ only in: (i) the pressure transmitting medium, (ii) the measurement method, and (iii) whether samples were first triaged based on transport measurements. Thus, our overall understanding from synthesizing the literature and from discussions with Meng is that the measured superconducting volume fraction can depend quite drastically on whether one utilizes an optimal pressure transmitting medium (such as helium gas) and also on whether one pre-selects samples based on “better” transport results. We have added these important discussions and context to our revised manuscript.

2) Could the authors make any statement on the sample alignment relative to the compression axis? Perhaps the authors could determine the crystal orientation after recovery from the high pressure cell, for example, of the sample shown in Extended Data fig. 4d? If the σ_{zz} and τ can be presented relative to the crystal axes, the phase diagram would be even more informative.

The referee’s suggestion is of course excellent! Knowledge about the crystal orientation would indeed significantly add to the informational content of our phase diagram. To be maximally honest, we had definitely been meaning to do single-crystal x-ray diffraction on our nickelate samples and in fact had already applied for beamtime at HPCAT before the submission of our manuscript. Motivated by this referee’s question and by the timing of the beamtime during the review process, we have now indeed taken single-crystal x-ray diffraction measurements (which is part of the reason our reply has taken some time), which we describe in detail below!

We begin with a few general remarks about the sample preparation: We find that, when cleaving the samples, they divide preferentially into thin slices of material. Moreover, along these plate-like samples, the crystal surfaces are noticeably smooth and shiny (see Fig. R5(b)). As this appears to be a “fast axis” upon cleaving, it is suggestive that the a and b crystalline axes lie in this plane, because the $\text{La}_3\text{Ni}_2\text{O}_7$ crystal structure is formed of layers in the a-b plane stacked along the c axis. As suggested by the referee, to further provide evidence for this, we have performed single-crystal X-ray diffraction (at HPCAT at Argonne National Laboratory) on a separate $\text{La}_3\text{Ni}_2\text{O}_7$ sample,

FIG. R5: **a** Single crystal x-ray diffraction pattern of $\text{La}_3\text{Ni}_2\text{O}_7$. The peaks which feature some contribution from the pressure transmitting medium, KBr, are indicated in blue underneath the experimental and calculated curves. The 2θ at which the (002) peak is expected to appear is noted with a grey dashed line, as it is otherwise absent in the diffraction pattern. Other prominent peaks from the $\text{La}_3\text{Ni}_2\text{O}_7$ sample are also labeled, though the (200), (020), and (220) peaks most clearly and simply allude to the orientation of the sample. **b** Scanning electron microscope (SEM) image of a typical cleaved $\text{La}_3\text{Ni}_2\text{O}_7$ sample. The sample is clearly much thinner than it is wide, and its surface is noticeably smooth.

which has been prepared in a DAC up to 5 GPa with an identical procedure. The X-ray diffraction data was integrated in Dioptas and subsequently processed via Rietveld refinement in GSAS-II to produce a calculated profile including the structure for the sample ($\text{La}_3\text{Ni}_2\text{O}_7$) [17] and the pressure transmitting medium (KBr). Upon refinement (see Fig. R5(a)), we find that the most prominent peaks are those which correspond to Bragg planes (200), (020), and (220). This, along with the clear lack of a peak corresponding to the (002) plane, suggests that the incident X-rays, and hence the **loading axis, is collinear with the c-axis of the crystal**. In particular, the small 2θ value of (200), (020), and (002) suggests they are only visible when the incident beam is nearly parallel to them. Thus, the incident beam is nearly parallel to both (200) and (020), which correspond to the crystalline b-c and a-c planes, but is certainly not parallel to the a-b plane. Thus, the beam is along the c axis.

3) The ability of resolving the shear components from the total stress tensor is one of the highlights of this paper. However, the correlation between superconductivity and shear stress is not so clear. Why, in Fig. 4h, does the optimal superconductivity appear at non-zero shear stress? The left axis of the panel appears cut off at around 0.4 GPa too. Are there no data points near 0 shear stress?

How is the panel f constructed then?

We appreciate this referee’s keen observation and agree that this is an intriguing point. Indeed, at such high pressures (~ 20 GPa), finding a shear stress near zero is actually exceedingly rare. The reason is that shear stresses are fundamentally associated with the contact friction between the sample and the diamond; thus, one always expects some finite magnitude, but of course, at lower overall pressures the magnitude of the shear stress will also be quite low. At our optimal pressures of a few 10s of GPa, we find that spatial points with $\tau < 0.4$ GPa are quite rare and thus we cut off the y-axis around there.

The fact that the data seem to exhibit a finite optimal shear for superconductivity is indeed a very intriguing feature. We emphasize that for Fig. 4h (where one sees the optimal shear), there is no interpolation of the data, so in that sense the optimal shear feature absolutely exists in the raw data set. Of course, there is always the experimental question of whether we have a sufficient number of samples to make this claim robustly. With this in mind, we do not make any claims about an optimal shear value for the superconducting response of $\text{La}_3\text{Ni}_2\text{O}_7$. In some sense, we share the referee’s intuition that it would perhaps be most natural for the optimum to occur at $\tau = 0$ GPa; however, after consulting with a number of theorists, it seems that this a priori intuition is not always perfect and there are materials where superconductivity is in fact enhanced by shear [18]. To this end, one of the major hopes that we have for our work, is that it motivates some careful *ab initio* calculations, which clarify the microscopic role of shear stresses in nickelate superconductivity.

The referee bring up a sharp question about Fig. 4f. We apologize for the confusion there and should have been more careful in explaining our schematic 3D phase diagram. Since we do not have data points down to $T = 0$ K, $\tau = 0$ GPa, or $\sigma_{zz} = 0$ GPa, in order to show a phase diagram that spans the full axes range, we construct panel 4f by extrapolating our data down to zero on all three axes using a spline fit (explained in detail in the Methods). To avoid any further confusion, we have now added an explicit sentence to the caption describing this.

If one examines the correlation between Fig. 4d and e, there are spots colored in bright yellow in panel e that appears superconducting too. This also leads to the question of why the color scales for panel g and h are not set at max of 1.

We appreciate the referee’s keen eye and excellent question! Indeed, the referee is absolutely correct

FIG. R6: Superconductivity phase diagram in the stoichiometry temperature plane. We note that the continuous nature of the La:Ni ratio axis owes to the finite spatial resolution of EDX, which may average over different chemical compositions. From the phase diagram, one observes a superconducting dome centered around an optimum La:Ni ratio of 1.5 where a maximum transition temperature of approximately 60K is seen.

that there are a few scattered points where spots supporting high shear also appear to superconduct. Our determination of the critical τ comes from averaging over all pixels, at all pressures, for all samples; to this end, we are, in principle, not surprised by some small variance around the otherwise relatively sharp threshold. This variance could come from a number of sources, including variance in the determination of the local shear stress, variance in the measurement of the local diamagnetic suppression, variance in the local chemical environment, etc. Finally, we respectfully emphasize that the visual correlation (see in Fig. 4d,e) and the dome-shaped phase diagram derived from the correlation is quite robust; with this in mind, one might say that the outlying yellow points are exceptions that prove the trend.

We thank the referee for their question regarding the choice of color scales. To be maximally honest, the original color bar limits were chosen to avoid some visual over-saturation of “blue” in certain regions of the phase diagram. However, we completely agree with the referee that choosing such a color scale can be quite confusing for the reader and we have now replotted both phase diagrams (Fig. 4g,h) using the same color bar and setting the maximum to 1.

4) Since the stoichiometry map is also 2D, could the authors construct a similar stoichiometry temperature phase diagram? It is rather strange how Fig. 5c only contains 8 data points with zero stoichiometry error bar. It would appear that a high-statistics statement on stoichiometry is possible here as well

We thank the referee for this wonderful suggestion. Motivated by the referee’s point, we have now constructed a 2D stoichiometry/temperature phase diagram, as shown in Fig. R6. As expected, this phase diagram reproduces the two qualitative features observed in the main text: (i) $T_c \approx 60\text{K}$ and (ii) an optimal La/Ni ratio ≈ 1.5 , but also provides significantly more quantitative information in the temperature-stoichiometry plane. We have now included this phase diagram in our Extended Data Fig. 4e. Finally, let us respond to the referee’s question about stoichiometry error bar; in Fig. 5c, we bin data along the stoichiometry axis in increments of 0.2 and then average the B/H for all points within the bin, hence the lack of ‘error bar’ on the stoichiometry axis.

Some minor questions: 1) In Fig.4d, at some locations of sample S1, a maximum of $> 6\%$ diamagnetic signal is observed. However, in Fig.4g and h, the largest suppression is 3.5% . Why is there a discrepancy?

We thank the referee again for their sharp observation. The referee is absolutely correct that the maximum diamagnetic suppression signal is certainly greater than 3.5% in sample S1. However, our phase diagrams are constructed by combining the data across all pressures, temperatures and samples, which generally exhibit suppression signals less than 3.5% . Thus, if one extends the color map to near the maximal diamagnetic suppression value, this leads an extremely “washed out” coloring of the phase diagram (see comparison shown in Fig. R7). At the time, we did try to play around with non-linear color maps to help with this issue, but ultimately decided that a linear color map seemed the simplest and most natural to use.

2) Regarding statistics and error bars, Fig. 5c suggests that there are appreciable errors in the B/H value which is not reflected in other figures, such as Fig. 5b, 3c, 2d.

We again thank the referee for their attention to this detail. We respectfully emphasize that the error bars in Fig. 5c are associated with the binning of data along the stoichiometry axis in increments of 0.2, and these error bars are on the order of ± 0.0005 . This is roughly one to two orders of magnitude smaller than the variations in B/H as a function of temperature, shown in Fig. 5b, 3c, 2d. In addition, the dominant error bar in the values of B/H (Fig. 5b, 3c, 2d.) arise from the statistical fluctuations associated with estimating the NV’s magnetic resonance line. Since we are able to resolve shifts in the NV’s ODMR spectrum at the level of $\sim 100\text{ kHz}$ ($\approx 0.04\text{ G}$), these uncertainties are smaller than the size of the markers in the plots.

3) This previous high pressure NV center work should be recognized and briefly discussed:

FIG. R7: Comparison of two phase diagrams with different color bars. When the color extends to account for the maximum diamagnetic suppression signals observed, this leads to a washed-out phase diagram where structure (in the Meissner suppression signal) near the boundaries of the transition region are difficult to resolve.

arxiv:2410.10275.

We completely agree with the referee and have now added a citation and discussion of this nice work to our main text, which has recently been published in NSR. We note that in addition to the reference mentioned by the referee, we found another related manuscript [Liu et al., PRL 135, 096001 (2025)], which confirms the presence of superconductivity using NV center measurements. As the referee mentions earlier on, neither of these studies perform widefield imaging, or demonstrate correlations with either local stress or chemical composition.

We thank the referee for their support and extremely careful reading of our manuscript. The referee's excellent suggestions have led to numerous modifications, which we believe have significantly strengthened our paper.

-
- [1] J.-Y. Ge, V. N. Gladilin, N. E. Sluchanko, A. Lyashenko, V. B. Filipov, J. O. Indekeu, and V. V. Moshchalkov, *New Journal of Physics* **19**, 093020 (2017).
- [2] R. M. da Silva, M. V. Milošević, A. A. Shanenko, F. M. Peeters, and J. A. Aguiar, *Scientific Reports* **5**, 12695 (2015).
- [3] A. Eyal and G. Koren, "Experimental evidence of T_c enhancement above 50 K and diode and paramagnetic-Meissner effects, in Nickelate films on highly reduced SrTiO_3 ," (2025).
- [4] W. Braunisch, N. Knauf, G. Bauer, A. Kock, A. Becker, B. Freitag, A. Grütz, V. Kataev, S. Neuhausen, B. Roden, D. Khomskii, D. Wohlleben, J. Bock, and E. Preisler, *Physical Review B* **48**, 4030 (1993), publisher: American Physical Society.
- [5] M. R. Koblishka, L. Püst, C.-S. Chang, T. Hauet, and A. Koblishka-Veneva, *Metals* **13**, 1140 (2023), publisher: Multi-disciplinary Digital Publishing Institute.
- [6] M. Sgrist and T. M. Rice, *Rev. Mod. Phys.* **67**, 503 (1995).
- [7] Y. Zhou, J. Guo, S. Cai, H. Sun, C. Li, J. Zhao, P. Wang, J. Han, X. Chen, Y. Chen, Q. Wu, Y. Ding, T. Xiang, H.-k.

- Mao, and L. Sun, *Matter and Radiation at Extremes* **10**, 027801 (2025).
- [8] P. Kehayias, M. Turner, R. Trubko, J. Schloss, C. Hart, M. Wesson, D. Glenn, and R. Walsworth, *Physical Review B* **100**, 174103 (2019).
- [9] D. A. Broadway, B. Johnson, M. Barson, S. E. Lillie, N. Dontschuk, D. McCloskey, A. Tsai, T. Teraji, D. Simpson, A. Stacey, *et al.*, *Nano Letters* **19**, 4543 (2019).
- [10] T. Tsuji, S. Harada, and T. Teraji, *Science and Technology of Advanced Materials* **26**, 2546779 (2025).
- [11] H. Sun, M. Huo, X. Hu, J. Li, Z. Liu, Y. Han, L. Tang, Z. Mao, P. Yang, B. Wang, J. Cheng, D.-X. Yao, G.-M. Zhang, and M. Wang, *Nature* **621**, 493 (2023).
- [12] J. Li, D. Peng, P. Ma, H. Zhang, Z. Xing, X. Huang, C. Huang, M. Huo, D. Hu, Z. Dong, X. Chen, T. Xie, H. Dong, H. Sun, Q. Zeng, H.-k. Mao, and M. Wang, *National Science Review*, nwaf220 (2025).
- [13] J. Hou, P.-T. Yang, Z.-Y. Liu, J.-Y. Li, P.-F. Shan, L. Ma, G. Wang, N.-N. Wang, H.-Z. Guo, J.-P. Sun, Y. Uwatoko, M. Wang, G.-M. Zhang, B.-S. Wang, and J.-G. Cheng, *Chinese Physics Letters* **40**, 117302 (2023).
- [14] Y. Zhang, D. Su, Y. Huang, Z. Shan, H. Sun, M. Huo, K. Ye, J. Zhang, Z. Yang, Y. Xu, Y. Su, R. Li, M. Smidman, M. Wang, L. Jiao, and H. Yuan, *Nature Physics* **20**, 1269 (2024).
- [15] G. Wang, N. Wang, X. Shen, J. Hou, L. Ma, L. Shi, Z. Ren, Y. Gu, H. Ma, P. Yang, *et al.*, *Physical Review X* **14**, 011040 (2024).
- [16] Y. Zhou, J. Guo, S. Cai, H. Sun, C. Li, J. Zhao, P. Wang, J. Han, X. Chen, Y. Chen, *et al.*, *Matter and Radiation at Extremes* **10** (2025).
- [17] X. Chen, J. Zhang, A. S. Thind, S. Sharma, H. LaBollita, G. Peterson, H. Zheng, D. P. Phelan, A. S. Botana, R. F. Klie, *et al.*, *Journal of the American Chemical Society* **146**, 3640 (2024).
- [18] M. Mito, H. Matsui, K. Tsuruta, T. Yamaguchi, K. Nakamura, H. Deguchi, N. Shirakawa, H. Adachi, T. Yamasaki, H. Iwaoka, Y. Ikoma, and Z. Horita, *Scientific Reports* **6**, 36337 (2016), publisher: Nature Publishing Group.